# Unraveling the influences of sequence and position on yeast uORF activity using massively parallel reporter systems and machine learning

Gemma E May[1], Christina Akirtava[1], Matthew Agar-Johnson[1], Jelena Micic[1], John Woolford[1], Joel McManus[1,2]*

[1]Department of Biological Sciences, Carnegie Mellon University, Pittsburgh, United States; [2]Computational Biology Department, Carnegie Mellon University, Pittsburgh, United States

**Abstract** Upstream open-reading frames (uORFs) are potent *cis*-acting regulators of mRNA translation and nonsense-mediated decay (NMD). While both AUG- and non-AUG initiated uORFs are ubiquitous in ribosome profiling studies, few uORFs have been experimentally tested. Consequently, the relative influences of sequence, structural, and positional features on uORF activity have not been determined. We quantified thousands of yeast uORFs using massively parallel reporter assays in wildtype and Δ*upf1* yeast. While nearly all AUG uORFs were robust repressors, most non-AUG uORFs had relatively weak impacts on expression. Machine learning regression modeling revealed that both uORF sequences and locations within transcript leaders predict their effect on gene expression. Indeed, alternative transcription start sites highly influenced uORF activity. These results define the scope of natural uORF activity, identify features associated with translational repression and NMD, and suggest that the locations of uORFs in transcript leaders are nearly as predictive as uORF sequences.

*For correspondence:
mcmanus@andrew.cmu.edu

## Editor's evaluation

This important study advances our understanding of how upstream reading frames contribute to gene expression regulation. Using innovative tools, the authors provide convincing evidence connecting the features of these sequences to protein expression. The results will be of broad interest to investigators in the field of gene expression regulation and its evolution.

## Introduction

The synthesis of cellular proteins by mRNA translation is an essential process regulated by multiple interactions between *cis*-acting sequences and *trans*-acting factors. Translation initiation is highly regulated to control the rate of protein synthesis. During canonical 5′ cap-dependent initiation, pre-initiation complexes (PICs) assemble at mRNA 5′ ends and scan directionally in search of start codons (*Hinnebusch et al., 2016*). Due to this directional scanning, mRNA sequences and structures in 5′ transcript leaders have a direct impact on initiation frequency. In particular, upstream Open Reading Frames (uORFs), short coding sequences between the 5′ cap and primary protein coding sequence (CDS), generally decrease the frequency of initiation at downstream CDSs (*Wethmar, 2014*). Termination at uORF stop codons can also induce nonsense mediated decay (NMD), a major mRNA turnover

pathway. As a result, most uORFs are expected to reduce gene expression. Despite this general view, the extent of uORF repression can vary greatly and the underlying causes for this remain unclear.

Despite their general repressive nature, some uORFs enhance expression of their downstream CDS, especially in response to stress (*Young and Wek, 2016*). A classic example of a gene harboring enhancer uORFs is yeast GCN4, whose transcript leader harbors four uORFs (*Mueller and Hinnebusch, 1986*). After translation of uORF1, the small subunit of the ribosome remains attached to *GCN4* mRNA, reforming a PIC that subsequently resumes scanning. In the absence of stress, the resulting PIC is rapidly recharged with tRNA-Met in a ternary complex with eIF2 and GTP, which leads to translation of uORFs 2–4 and a corresponding inhibition of *GCN4* CDS translation. However, most stress conditions cause phosphorylation of the eIF2α subunit, which allows rescanning PICs to scan past uORFs 2–4 to translate the *GCN4* CDS (*Hinnebusch, 2005*). A similar multi-uORF reinitiation system allows stress-dependent translation of mammalian ATF4, in which resumption of scanning after translating a short uORF allows subsequent leaky scanning past a more repressive uORF (*Vattem and Wek, 2004*). In theory, the same mechanism could allow short uORFs to insulate against longer, more repressive uORFs even in the absence of stress (*Lin et al., 2019*). However, the frequency of such stress-independent uORF enhancers remains unknown.

Although once thought to be relatively rare, genomic studies have revealed that AUG-initiated uORFs are common, affecting ~15% of yeast (*Ingolia et al., 2009*; *McManus et al., 2014*; *Zhang and Dietrich, 2005a*) and ~50% of human genes (*McGillivray et al., 2018*; *Wethmar et al., 2014*). In addition to these canonical elements, ribosome profiling studies using drugs that stall initiating ribosomes, have identified even more uORFs that initiate at non-AUG codons (*Brar et al., 2011*; *Ingolia et al., 2011*; *Spealman et al., 2018*). However, other work suggests the drugs used in those studies may exaggerate the frequency of uORF usage (*Gerashchenko and Gladyshev, 2014*; *Kearse et al., 2019*; *Lareau et al., 2014*). Despite the large number of predicted AUG and non-AUG uORFs, most studies focus solely on identifying these elements and few have been experimentally tested (*Wethmar et al., 2014*). The few non-AUG uORFs that have been tested had relatively modest influences on expression (*Spealman et al., 2018*; *Zhang and Hinnebusch, 2011*). Thus, the relative influence of AUG- and non-AUG uORFs on mRNA translation has not been systematically evaluated.

Due to their frequency and functional importance, uORF evolution has been the subject of multiple studies (*Zhang et al., 2019*). Early work identified 38 genes with AUG-initiated uORFs that were deeply conserved among yeast species by stringent comparisons of draft genome sequences (*Zhang and Dietrich, 2005a*). Functional evaluations of nine *S. cerevisiae* uORFs showed six altered the expression in a luciferase reporter assay. More recently, genome-wide studies using ribosome profiling identified thousands of AUG and non-AUG uORFs in multiple *Saccharomyces* species, some of which were translated in multiple species (*Spealman et al., 2018*). Ribosome profiling was also used to identify uORFs used throughout *Drosophila* development, many of which appear to be adaptive due to signs of positive selection detected by comparative genomics (*Zhang et al., 2018*). Additional analyses of human, mouse, and zebrafish ribosome profiling data estimated uORF regulatory activity using the ratio (U/C) of ribosome footprints in 5' transcript leaders (UTRs) to footprints in coding sequences (CDS) (*Chew et al., 2016*; *Johnstone et al., 2016*). These studies found modest, though statistically significant correlations of U/C ratios in data from mouse and human tissue culture cells and bulk brain tissue, suggesting that uORF activities are somewhat conserved in vertebrates. However, to our knowledge, the regulatory functions of homologous uORFs from different species have not been experimentally compared. Thus, the extent to which uORF functions are quantitatively conserved across species remains unclear.

Currently, the magnitude of uORF activity is thought to depend primarily on the extent to which uORF start codons match the Kozak consensus sequence (*Cuperus et al., 2017*; *Dvir et al., 2013*; *Noderer et al., 2014*; *Sample et al., 2019*). Analyses of ribosome profiling data from vertebrates found that the U/C ratio was higher for uORFs with start codons in strong Kozak contexts (*Chew et al., 2016*; *Johnstone et al., 2016*). Massively parallel reporter assays (MPRAs) of random transcript leaders confirmed that AUG-uORFs in strong Kozak contexts are repressive (*Cuperus et al., 2017*; *Dvir et al., 2013*; *Ferreira et al., 2014*; *Noderer et al., 2014*; *Sample et al., 2019*). However, other features have also been shown to affect uORF activity. For example, a uORF-specific MPRA (FACS-uORF) study found rare codons and dicodons within a uORF can determine whether the uORF enhances or represses expression of the CDS (*Lin et al., 2019*). The structural accessibility of start

codons was also found to play a key role in determining the impact of uORFs from the human α–1-antitrypsin mRNA leader (*Corley et al., 2017*). Other work has shown that uORF activity can depend on the location of uORFs relative to main protein-coding ORF start codons (*Beznosková et al., 2013*; *Grant et al., 2012*). However, the relative influences of these features on natural uORF activities have not been determined, primarily because few such uORFs have been functionally assayed.

Similarly, many questions remain regarding uORFs and NMD. The extent to which NMD contributes to typical uORF-mediated gene repression remain unclear, as do the roles of uORF features in determining NMD susceptibility. While uORFs are clearly correlated with NMD (*Celik et al., 2017*; *Smith et al., 2014*), the yeast GCN4 and YAP1 uORFs appear to resist NMD (*Ruiz-Echevarría and Peltz, 2000*). Early studies suggested yeast NMD requires AU-rich *cis*-acting downstream sequence elements bound by Hrp1p (*Culbertson and Leeds, 2003*; *Czaplinski et al., 1999*; *González et al., 2000*; *Peltz et al., 1993*; *Ruiz-Echevarría et al., 1998*); however, such elements are not well defined and appear to be missing from many NMD targets (*Meaux et al., 2008*). Other work found premature termination codons (PTCs) were less likely to induce NMD when inserted closer to the 5' end of a yeast PGK1 mRNA (*Muhlrad and Parker, 1999*). NMD induction was also shown to be greatly reduced by low-frequency stop codon readthrough at PTCs (*Keeling et al., 2004*); however, high-frequency readthrough did not prevent NMD in another context (*Gorgoni et al., 2019*). Other studies showed mRNA can be protected from NMD by positioning poly(A)-binding protein close to the termination codon (*Amrani et al., 2004*; *Eberle et al., 2008*). Additional work in human cells found translation initiation factors inhibit NMD, as does reinitiation after termination at PTCs (*Lindeboom et al., 2016*; *Raimondeau et al., 2018*; *Zhang and Maquat, 1997*). While these studies helped explain NMD induction by PTCs in coding genes, most did not consider uORFs, which may behave differently due to their extreme 5' locations in transcript leaders and generally short lengths. Thus, the features that influence uORF induction of NMD remain unclear. Furthermore, the relative importance of NMD and translation inhibition in natural uORF activity has not been systematically evaluated.

Here, we used two MPRA systems, FACS-uORF and PoLib-seq, to quantify the impact of thousands of natural yeast AUG- and non-AUG uORFs on protein expression and ribosome loading. Our results show that most non-AUG uORFs have small impacts on expression compared to AUG-uORFs. Leveraging the massive scale of our results, we evaluated the influence of sequence and positional features of natural AUG-uORFs on their gene regulatory effects. Using a strain deleted of the NMD factor *UPF1* (Δ*upf1*), we showed NMD accounts for roughly a third of uORF repression in yeast, and further investigated how uORF features impact the propensity for NMD. Finally, we used elastic-net regression modeling to select and weight features that correlate with uORF activity, revealing that uORF locations are as predictive for uORF function as Kozak contexts. Surprisingly, we found uORF activity often depends on the site of transcription initiation, as alternative transcription start sites can dramatically change the magnitude of uORF repression.

## Results

### FACS-uORF determines the effects of thousands of AUG and non-AUG uORFs on protein expression

To evaluate the impacts of natural yeast uORFs on gene expression, we used FACS-uORF (*Lin et al., 2019*; *Figure 1A*). FACS-uORF simultaneously compares YFP expression from thousands of wildtype uORFs in endogenous transcript leaders to corresponding mutants in which the predicted uORF start codon has been mutated to a non-functional AAG. Our reporter library design included all transcript leaders less than 180 nucleotides long that contain at least one uAUG from *S. cerevisiae* (1,524 uORFs) and *S. paradoxus* (1,206 uORFs), as well as all *S. cerevisiae* non-AUG uORFs (540 uORFs) that we previously identified using ribosome profiling (*Spealman et al., 2018*). Start codons were individually mutated in transcript leaders containing multiple uORFs. After removing low-frequency plasmid constructs with highly variable YFP levels (Methods, *Figure 1—figure supplement 1*), 1,689 unique AUG and 349 non-AUG uORFs remained with confident measurements of activity. Importantly, we verified that transcription initiated at the designed positions for ~97% of reporter transcripts using targeted 5' CAGE-seq analysis (*Figure 1—figure supplement 2*), and that measurements using mCherry as the reporter were highly similar (*Figure 1—figure supplement 3*). To our knowledge, this represents the largest panel of natural uORFs assayed to date.

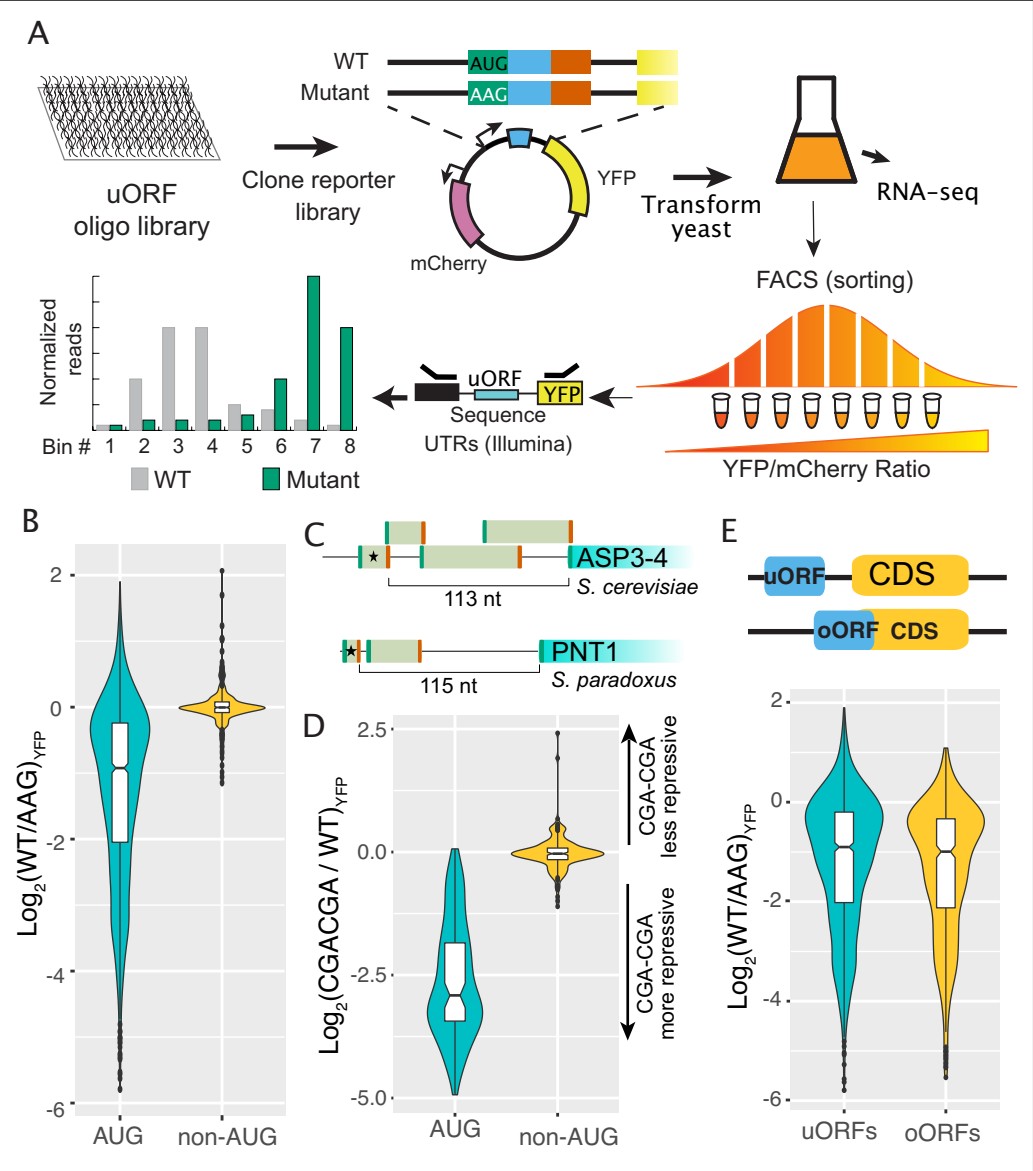

**Figure 1.** Regulatory impacts of 2038 yeast uORFs. (**A**) A custom library of pairs of uORF-containing and uORF-mutant transcript leaders was cloned into reporter plasmids. Yeast were transformed with the reporter library and FACS sorted on the YFP / mCherry ratio. Plasmids extracted from the resulting FACS bins were sequenced to measure the expression levels of each reporter. (**B**) Comparison of AUG- and non-AUG-uORF activities. The log-fold change resulting from mutating uORF start codons is plotted. AUG uORFs are much more repressive than NCC-uORFs, though some AUG uORFs enhance expression. (**C**) Examples of complex transcript leaders with two enhancer uORFs (stars). (**D**) Insertion of CGACGA stalling dicodons increases repression from all AUG-uORFs and most non-AUG-uORFs, supporting their translation. (**E**) Overlapping AUG-uORFs (oORFs) are slightly more repressive than discrete uORFs.

The online version of this article includes the following figure supplement(s) for figure 1:

**Figure supplement 1.** Comparison of UTR expression measurements using FACS.

**Figure supplement 2.** Fidelity of transcription initiation in FACS-uORF reporter constructs.

**Figure supplement 3.** Fluorescent protein identity has minor impacts on measurement of uORF functions.

**Figure supplement 4.** Examples of false-positive uORF enhancers.

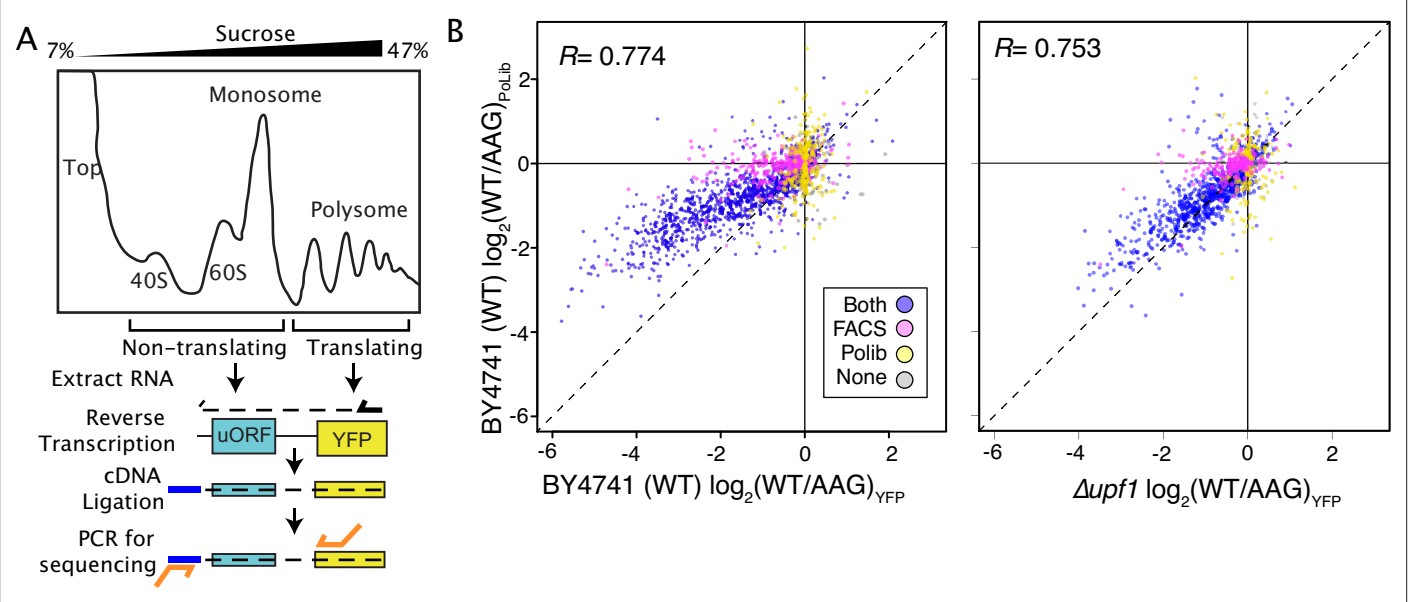

**Figure 2.** Massively parallel analysis of ribosome loading by Polysome Library Sequencing (PoLib-seq) supports FACS-uORF estimates of uORF function. (**A**) The PoLib-seq assay system was developed to determine the impact of uORFs on ribosome loading. Polysome extracts were prepared from wildtype (BY4741) yeast strains expressing FACS-uORF library 2 and separated on a 7–47% sucrose gradient by untracentrifugation. RNA was extracted from polysomal (2+ribosome) and non-polysomal (40 S, 60 S, and monosome) fractions, and reporter constructs were quantified by targeted RNA-seq. The percent of polysomal reads was compared for each wildtype / AAG mutant uORF pair to determine the impact of each uORF on ribosome loading. (**B**) Comparison of FACS-uORF (x axis) and PoLib-seq (y axis) estimates of uORF function, both given as the log-transformed difference between wildtype and AAG mutant uORF. uORF impacts on ribosome loading are positively correlated with impacts on protein expression in wildtype yeast (left), and the correlation is stronger in the upf1Δ strain (right).

The online version of this article includes the following figure supplement(s) for figure 2:

**Figure supplement 1.** Reproducibility of PoLib-seq estimates of translation efficiency and uORF activity.

**Figure supplement 2.** Distribution of PoLib-seq reads.

We first evaluated the effects of AUG and non-AUG uORFs on reporter protein expression (*Figure 1*; *Supplementary file 1a*). Most (1222 of 1689; 72%) AUG-initiated uORFs significantly altered YFP expression (*Figure 1B*). The vast majority (95%) of these functional uORFs were repressors, causing a 2.8-fold median decrease in expression. We next evaluated the 66 enhancer AUG-uORFs, as uORFs with enhancer activities are uncommon. Most (37) could be explained by gene annotation errors or alternative mechanisms (*Figure 1—figure supplement 4*). For example, the two strongest "enhancer" uORFs were preceded by a U, such that the (U)A$\underline{U}$G ->A$\underline{A}$G mutation used created a UA$\underline{A}$ stop codon that converted unannotated N-terminal extensions into uORFs (Spar_9:277239–277245 and chrXV:423740-oORF). Several other enhancer uORFs had start codons preceding another AUG ($\underline{AUG}$NAUG, e.g. uORF chrXII:73376-73397), such that the upstream AUG ->AAG mutation places –3 A in the Kozak regions of the next AUG uORF, and are likely false positives resulting from activation of downstream uORFs (*Supplementary file 1b*). After removing suspected false positives from further study, the remaining 29 enhancer uORFs increased expression from 1.15-fold to 4-fold, with a median increase of 1.7-fold. Most (26) were in multi-uORF transcript leaders, such that their translation might alter the use of other uORFs. Indeed 12 enhancer uORFs were upstream of other uORFs, reminiscent to the *GCN4* uORF1 enhancer that insulates against initiation at repressive downstream uORFs under stress (e.g. *Figure 1C*). Thus, while most uORFs reduce gene expression, a small number act as enhancers, potentially by insulating against the effects of other, more repressive uORFs.

In comparison to AUG uORFs, non-AUG uORFs were less likely to change expression (215/349, 62%, Fisher's exact test (FET) p=8 x 10^–5) and had much smaller impacts, as mutating their start codons changed expression by fourfold or less (median 10% change, Wilcoxon Rank-sum Test (WRT) p=2.2 x 10$^{-16}$; *Figure 2A*). Non-AUG uORFs were eight times more likely to have enhancer activity than AUG-uORFs (102/215, 47%, FET p<2.2 × 10$^{-16}$). A manual evaluation of 30 non-AUG uORFs that

increased expression by at least 25% found 6 were located inside AUG uORFs, such that their mutation may alter the AUG uORF function. Similarly, 4 of 16 non-AUG uORFs that decreased expression by at least 25% were nested inside AUG uORFs. The location of these non-AUG uORFs inside AUG uORFs complicates interpretation of their functions by start codon mutation. In summary, these results indicate that non-AUG uORFs have milder impacts on expression, consistent with inefficient initiation at non-AUG codons.

We previously showed that insertion of the CGACGA dicodon, which stalls elongating ribosomes (*Letzring et al., 2013*), increased repression from the yeast *YAP1* uORF (*Lin et al., 2019*). We reasoned that CGACGA insertion should similarly increase repression of other translated uORFs. To test this, we used FACS-uORF to compare YFP expression from 100 significant AUG- and 164 non-AUG uORFs with and without CGACGA insertions (*Figure 1D*; *Supplementary file 1c*). The dicodon insertion made nearly all AUG-uORFs (99%), and most non-AUG uORFs (59%, Binomial Exact Test (BET) p=0.0235), more repressive. These results support the active translation of AUG- and most non-AUG uORFs, both enhancers and repressors. However, the significant regulatory effect of mutating non-AUG uORF start codons to AAG may not always result from a loss of translation initiation, as a stalling dicodon did not cause repression from a considerable fraction of non-AUG uORFs.

We next considered the regulatory impact of uORFs that overlap the main gene ORF (oORFs). Previous analyses of metazoan ribosome profiling data suggested that oORFs are more repressive than uORFs that terminate in transcript leaders (*Chew et al., 2016*; *Johnstone et al., 2016*). However, ribosome profiling may not accurately evaluate uORF regulatory effects because it captures noisy snapshots of ribosome occupancy over short sequence regions. By assaying the expression of wild-type and uORF-mutant reporter plasmids, we directly compared the regulatory effects of uORFs and oORFs in yeast. Considering all AUG-uORFs, we found that oORFs are ~10% more repressive than uORFs (*Figure 1E*; WRT p=0.02432), and this difference decreases when considering only AUG uORFs that have significant impacts on expression. Thus, our results show that oORFs are only slightly more repressive than uORFs in yeast.

## Consistent uORF impacts on protein levels and translation efficiency

By assaying steady-state protein levels, FACS-uORF measures aggregate changes in mRNA transcription, stability, and translation efficiency. As such, it is possible that the mutations used to inactivate uORF start codons might also affect reporter construct transcription and decay, rather than altering translation efficiency. This is a particular concern for apparent enhancer uORFs, which decrease YFP expression after start codon mutation. To validate the effects of uORFs on mRNA translation we developed a second MPRA for polysome loading, called Polysome Library sequencing (PoLib-seq *Figure 2A*; *Supplementary file 1d*). PoLib-seq involves sucrose gradient fractionation of polysome extracts from yeast carrying the reporter library, followed by directed RNA-sequencing to estimate the impact of uORFs on ribosome loading. PoLib-seq and FACS-uORF estimates were positively correlated, indicating general agreement between these two complimentary assays. Importantly, both enhancer uORFs and repressor uORFs identified from FACS-uORF had similar effects on ribosome loading as measured by PoLib-seq (*Figure 2B*). Of the 1,216 significant repressors of YFP found by FACS-uORF, 1,068 repressed ribosome loading in PoLib-seq measurements (88%, BET p<2.2 * $10^{-16}$). Similarly, of the 120 AUG and non-AUG FACS-uORF YFP enhancers, 86 increased ribosome loading in PoLib-seq (72% p<2.3 * $10^{-6}$). To further evaluate this, we performed FACS-uORF using a strain in which NMD is eliminated (*upf1Δ*). The slope of the linear regression between FACS-uORF and PoLib-seq results is closer to 1 in *upf1Δ* than in wildtype yeast (*Figure 2B*, p=3.6 * $10^{-31}$), indicating that PoLib-seq captures the translational effect of uORFs independent of NMD. Thus, the FACS-uORF and PoLib-seq results were generally consistent, underscoring the regulatory impact of these uORFs on mRNA translation. However, because PoLib-seq results were noisier than FACS-uORF results (*Figure 2—figure supplement 1*), we used FACS-uORF data for the remainder of this study.

## Dissecting the contribution of NMD in uORF repression

To examine the influence of NMD on uORF repression, we next compared uORF activity in wildtype and *upf1Δ* yeast (*Figure 3A*; *Supplementary file 1e*). In general, uORFs that reduced expression by 15% or more in wildtype yeast were *less* repressive in the *upf1Δ* strain, including 93% of AUG-uORFs (BET p<2.2 x $10^{-16}$) and 85% of non-AUG uORFs (BET p=0.0005335). The decrease in repression that

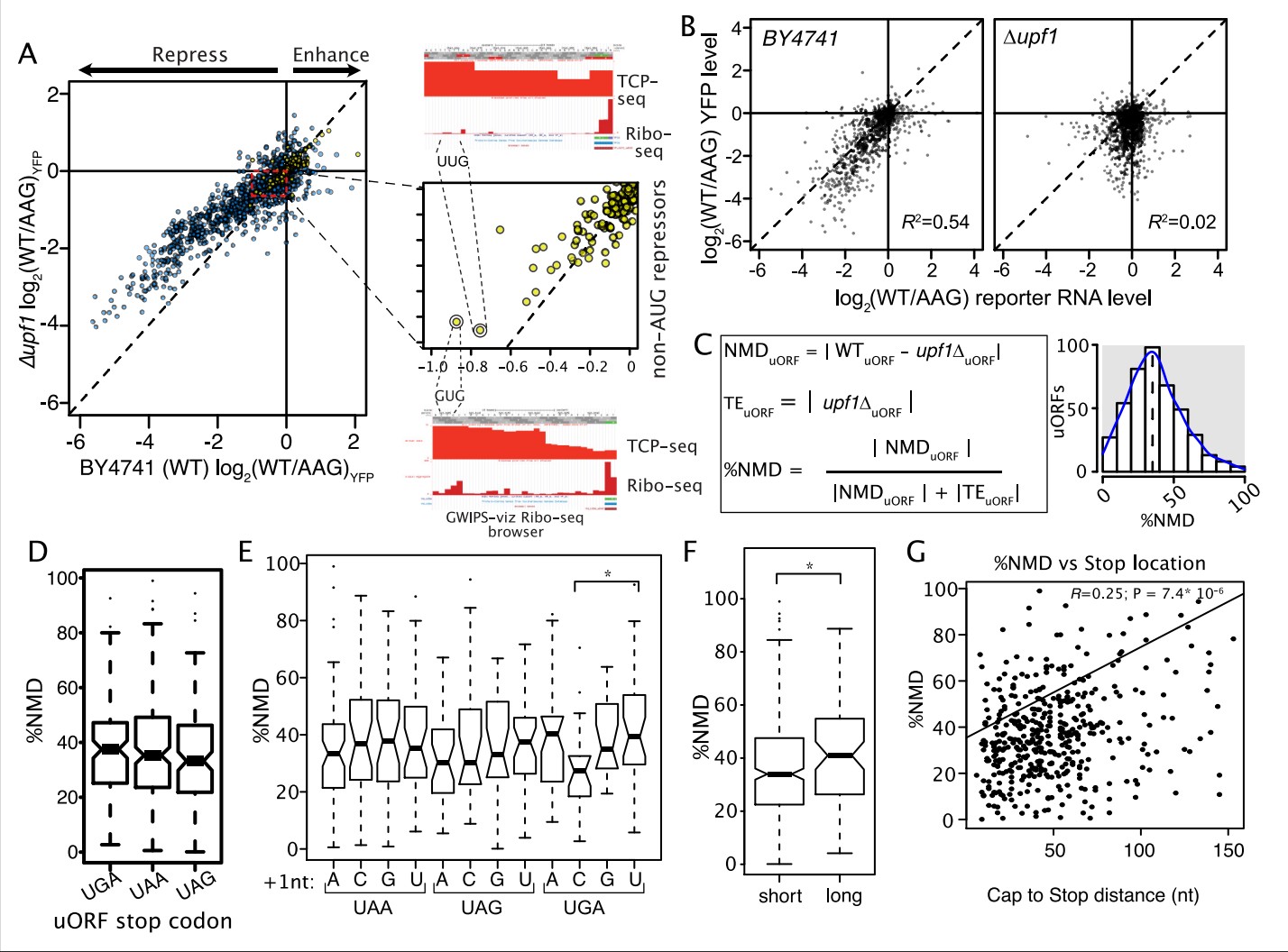

**Figure 3.** The role of nonsense-mediated decay in uORF activity. (**A**) Scatter plot compares the regulatory impact of mutating uORF start codons in wildtype (x-axis) and Δupf1 mutant yeast defective for nonsense mediated decay (left). Most AUG (blue), and some non-AUG uORFs (yellow) were less repressive in a Δupf1 strain, compared to wildtype. A zoomed view of non-AUG uORF NMD effects is shown, highlighting two relatively strong non-AUG-repressor uORFs. Composite Ribo-seq and TCP-seq profiles from the GWIPS-viz browser supporting translation of these uORFs are shown (right). (**B**) Comparison of uORF impacts on mRNA and protein expression levels are shown for wildtype (left) and Δupf1 (right) yeast strains. Δupf1 decouples uORF impacts on transcription and translation. (**C**) Calculating the percent of each AUG uORF's regulation due to NMD. WT$_{uORF}$ and $upf1\Delta_{uORF}$ are the log2(WT/AAG) in wildtype and $upf1\Delta$ yeast strains, respectively. The magnitude of uORF regulation through NMD (NMD$_{uORF}$) is calculated as WT minus $upf1\Delta$, $upf1\Delta$uORF is the magnitude of the uORF effect solely on translation efficiency (TE$_{uORF}$), in the absence of NMD. The relative contributions of NMD to uORF activity (%NMD) is calculated as the fraction of the total regulation attributable to NMD. %NMD is plotted for all assayed uORFs, showing a wide range of NMD induction. (D through G) Association of uORF features with %NMD. Termination at UGA stop codons induces more NMD than other stop codons (**D**), however UGAC stop codons, which are known to allow read-through, have lower %NMD (*; P = 0.061). Long uORFs induce more NMD than shorter uORFs, and uORFs that terminate far from transcript leader caps induce more NMD than those that terminate close to 5' caps (*; P = 0.011).

was observed among strong inhibitory non-AUG uORFs (*Figure 3A* inset) indicates that at least some non-AUG uORFs induce NMD. Since uORF-induced NMD directly impacts RNA abundance, we used targeted RNA-seq to compare the effects of uORFs on RNA levels. In wild-type yeast, 54% of the variance in uORF effects on YFP protein levels could be explained by differences in RNA abundance (*Figure 3B*). Notably, this correlation was essentially eliminated in $upf1\Delta$, suggesting uORF-induced NMD plays a prominent role in reporter repression. (*Figure 3B*). By comparing the magnitude of uORF repression in wildtype (both translation and NMD) and $upf1\Delta$ (translation only), we quantified the contribution of NMD (%NMD; *Figure 3C*) for 431 significant repressor AUG-uORFs for which data

were available from both strains. Across these repressive uORFs the median percentage of NMD is 35%. Thus, we estimate that roughly one third of yeast uORF repression is due to NMD.

Our dataset provides a unique opportunity to examine the features associated with variation in uORF induction of NMD. NMD is induced through inefficient translation termination, and termination efficiency varies among the three stop codons (UAA~UAG >> UGA; *Bonetti et al., 1995*). Previous work suggested the extent of NMD induced by PTCs in coding regions depended on the stop codon identity (*Keeling et al., 2004*). Consistent with this, we found median %NMD was higher for uORFs terminating with UGA (37.5%) than those terminating with UAA (35.3%) or UAG (33.2%; *Figure 3D*), though this was not statistically significant. The first nucleotide after the stop codon is known to influence termination through base-stacking interactions with rRNA (*Brown et al., 2015*). The %NMD varied among all 12 stop +1 sequences (*Figure 3E*). Although this was not generally significant (one-way ANOVA), the large variation among UGAN stop contexts approaches significance, with UGAC having lower %NMD (29.7%) than UGAU (41.3%; *Figure 3E*; Kruskal-Wallis test p=0.061). UGAC stop codons have been shown to allow stop codon read-through at rates higher than any other stop codon (*Cridge et al., 2018*; *Namy et al., 2001*). In summary, stop codon sequence context appears to influence the propensity for NMD by uORFs.

We reasoned that post-termination reinitiation at uORFs might also protect mRNA from NMD. As ribosomes more efficiently resume scanning after termination of short uORFs than long ones (*Gunišová et al., 2017*), we investigated the relationship between %NMD and uORF length. As expected, short uORFs ( ≤ 12 amino acids) exhibited lower %NMD than long uORFs (>12 amino acids) suggesting that termination leading to reinitiation reduces the propensity for NMD (*Figure 3F*; WRT p=0.011). Perhaps consistent with this, the location of the stop codon relative to the transcript leader cap was positively correlated with the %NMD, such that uORFs that terminate further from the cap were somewhat more likely to induce NMD than those that terminate adjacent to the cap (*Figure 3G*; $R^2$=0.065; p=7.44 × $10^{-8}$). Together, these results suggest that the propensity for uORFs to induce NMD depends on termination efficiency, uORF length, and, potentially uORF stop codon position in the mRNA transcript leader.

## Features predictive of uORF regulatory function

We next evaluated how sequence and structural features affect AUG and non-AUG uORF regulatory functions. The primary feature currently evaluated when examining uORF activity is the strength of the start codon Kozak context. In yeast, a strong Kozak context is typically rich in adenosine, particularly at the –3 position (*Li et al., 2019*). This characteristic 'strong' Kozak context was only observed among the six most repressive uORFs (32-fold or more repressors). It was entirely absent for uORFs that decreased expression less than 16-fold and for enhancer uORFs (*Figure 4A*). This suggests that other features may influence the activity of most yeast uORFs. Intriguingly, we found that the relative position of uORF start and stop codons correlated with differences in regulatory activity. Start codons located near the transcription start site were more repressive than those positioned further downstream, while stop codons further from the main ORF (in this case YFP) start codon were less repressive (*Figure 4B–C*). These results suggest uORF location plays a more prominent role in determining regulatory function than has been previously appreciated.

Because many features of uORF sequence and location can correlate with their impact on gene expression, their relative influence can be difficult to disentangle. For example, uORF Kozak contexts could vary with their location due to differences in G/C content near main ORF start codons. We next used a machine learning approach to select features that influence natural uORF activity independently and quantify their effects. Based on our results, and prior work, we examined fifteen uORF features, including Kozak context, folding energy around the start codon, uORF position in the transcript leader, codon usage and peptide charge, uORF length, stop codon identity and sequence context, and start and stop codon conservation. Kozak context strengths were determined using FACS-sorting of a library of 5′ UTRs of all variations surrounding AUG- and non-AUG start codons (–4 to +1; *Figure 4—figure supplement 1*; *Supplementary file 1f*). Given the size of our dataset, we chose elastic net regression (ENR) to select and weight features that contribute to a linear regression model of natural uORF activity (see Methods). We built ENR models of uORF activity from both wild-type and *upf1Δ* strains to further investigate the role of uORF features in translational repression and NMD.

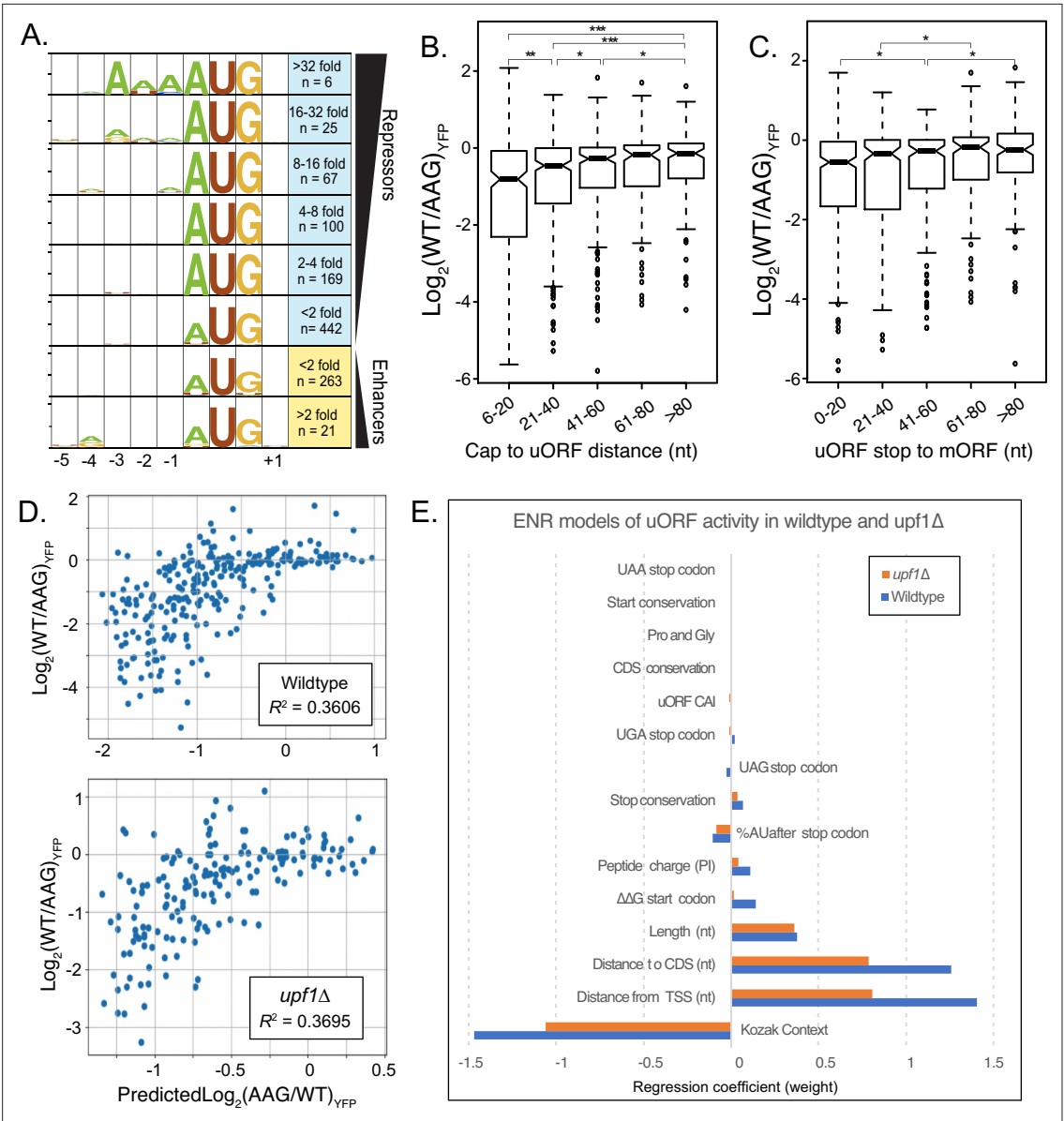

**Figure 4.** Elastic Net Regression modeling of uORF activities in wildtype and UPF1-delta strains. (**A**) Weblogo consensus sequences in the Kozak region surrounding uORF start codons, separated by regulatory effect. (**B**) Boxplot of regulatory effects for uORFs binned on distance between the 5' cap and the uORF start codon. uORFs starting before 6 nucleotides were removed to avoid the A-rich region immediately after the reporter transcription start site. (**C**) Boxplot of regulatory effects for uORFs binned by distance between the stop codon and the YFP start codon. (**D**) Scatterplot comparing uORF activities predicted by the elastic net regression model (x-axis) with those observed from FACS-uORF (y-axis) in wildtype yeast. The model explains roughly 1/3 the variance in uORF activity. (**E**) The bar graph shows feature weights in the ENR model. Notably, uORF start and stop codon location features are together more predictive than Kozak sequence strength. Similar results for ENR modeling of uORF activity in a upf1-deletion strain. In the absence of NMD, the sequence downstream of the stop codon is no longer significant, while the uORF codon Adaptation Index (uORF CAI) becomes significant. * indicates $P < 0.05$; ** shows $P < 0.0005$; and *** depicts ($P < 5 \times 10^{-8}$).

The online version of this article includes the following figure supplement(s) for figure 4:

**Figure supplement 1.** Determination of relative Kozak context strength for AUG and NCC codons using FACS-seq.

**Figure supplement 2.** Effects of uORF location on represion.

Many of the uORF features selected by the ENR models were similar for wildtype and *upf1Δ* strains. As expected, start codon context was the most predictive feature selected by the ENR in both, having a strong negative correlation with uORF activity. Surprisingly, two features describing uORF location – distance from the cap to the start codon and from the stop codon to the main ORF – were selected

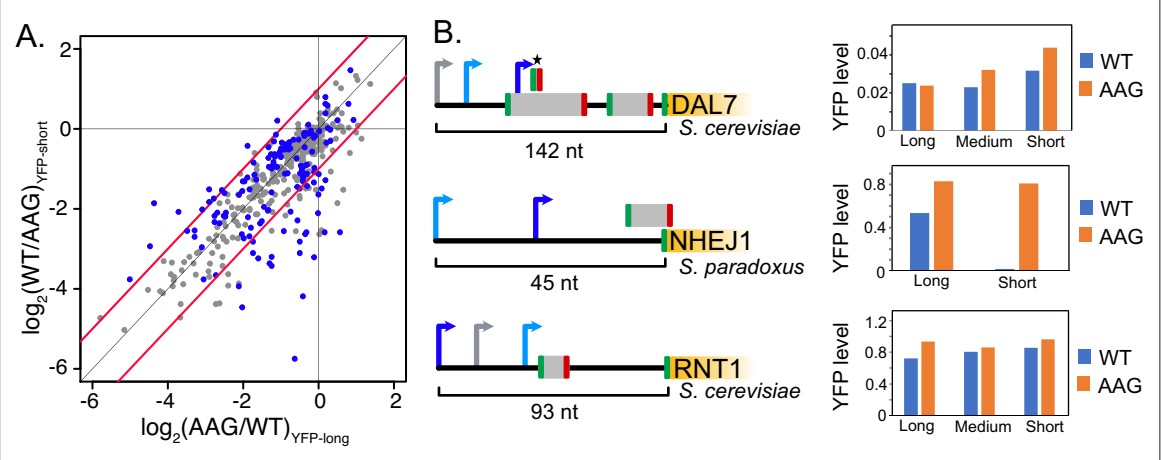

**Figure 5.** uORF activity depends on transcription start site usage. (**A**) Scatter plot comparing the activities of individual uORFs when located in longer (x axis) or shorter (y axis) transcript leaders resulting from alternative transcription start sites. Significant changes in uORF activity are indicated as red points in the plot (t-test adjusted p<0.05). Red lines mark twofold difference in activity. In most cases, uORFs are more repressive in shorter transcript leaders, when located closer to the transcription start site and 5' m7G cap. (**B**) Examples of uORFs with TSS location dependent activity. Arrows denote alternative transcription start sites. Blue arrows signify transcript leaders in which uORFs are more repressive, with dark blue indicating greater significance than light blue. Gray arrows indicate transcription start sites that were not significant. The asterisk shows the uORF whose repression varies with alternative transcription start site usage for DAL7. The bargraphs (right) show average YFP / mCherry levels for wild-type (WT) and uORF mutant (AAG) reporters for the transcript leaders indicated in the diagram.

independently and correlated with uORF strength nearly as strongly as did Kozak contexts (*Figure 5*). Both features had strong positive correlations with uORF activity, as uORFs initiating closer to the 5' cap, or terminating closer to the main ORF start codon, were more repressive. Notably, this trend was also observed in single-uORF transcript leaders (*Figure 4—figure supplement 2*). However, altering the cap-to-start distance of four uORFs (two cap-proximal and two cap-distal) did not always change uORF regulation as predicted by the ENR model (*Figure 4—figure supplement 2*). This suggests other features correlated to uORF location may have separate effects not captured by the model. Additionally, the four insertions and deletions tested alter not only the location, but also the sequence and structural context of uORF start codons. In either case, we conclude Kozak context and location are similarly predictive for uORF function.

uORF coding region features also contributed significantly, to the model, as shorter uORFs and uORFs encoding more negatively charged peptides were more repressive. However, these effects were relatively modest. ENR models from wildtype and *upf1Δ* strains were generally similar; however, the regression weights were generally greater in the wildtype ENR model. Together, the ENR models indicate that uORF Kozak context, position, length, and peptide charge are strong contributors to translational repression, while AU-rich downstream sequences may alter the propensity for NMD after uORF translation, consistent with historical reports of downstream sequence elements. Finally, because the ENR models explain only a third of the variation in natural uORF activity, other features or non-linear relationships likely have additional influence on uORF function.

## Variation in uORF functions in alternative transcript leader isoforms

Yeast use alternative transcription start sites that create corresponding alternative transcript leader sequences in response to environmental stimuli (*Arribere and Gilbert, 2013*; *Lu and Lin, 2019*; *Pelechano et al., 2013*). We compared the activities (WT / AAG) of 333 AUG and non-AUG uORFs in 470 alternative transcript leaders (*Supplementary file 1g*). Strikingly, 116 uORFs differed significantly depending on the transcription start site (t-test, p.adjust ≤ 0.05; *Figure 5A*). Of those that had at least a two-fold difference in activity, thirty four uORFs were more repressive when they were closer to the transcription start site, while only twelve were less repressive (BET *P*=0.001641). This result is consistent with our regression modeling, which identified uORF positions relative to 5' cap and main ORF start codon as important features that impact uORF function. For example, the uORF from the *S. paradoxus NEJ1* gene is ~60-fold repressive in the context of a transcript leader starting

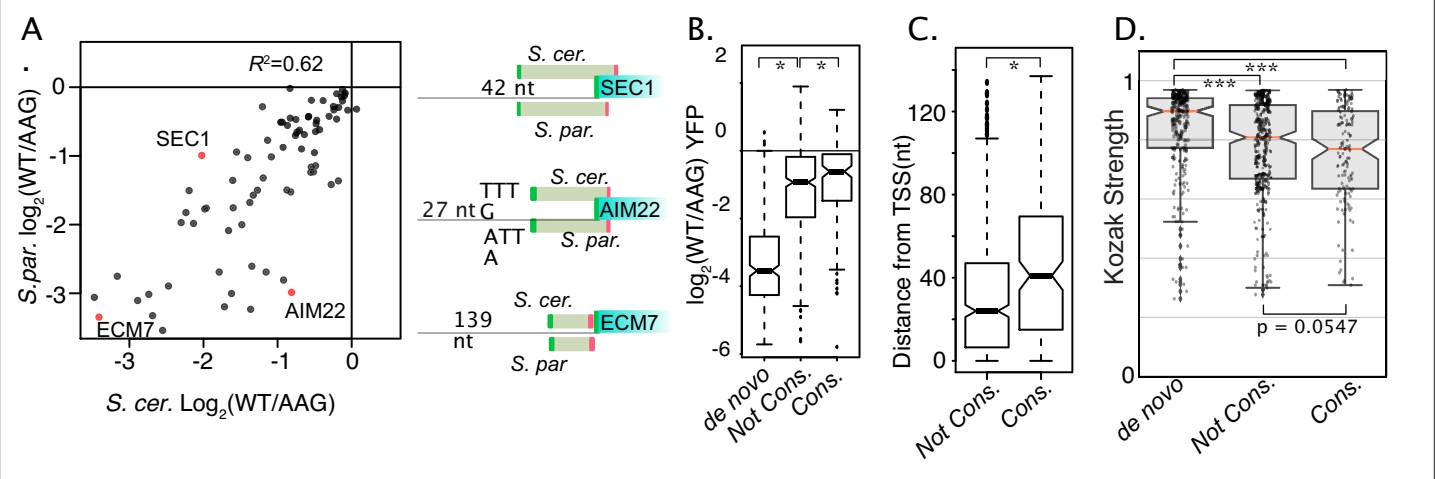

**Figure 6.** Evolutionary differences in uORF activities. (**A**) Scatter plot compares the activities of homologous uORFS from *S. cerevisiae* (x axis) and *S. paradoxus* (y axis) all assayed in *S. cerevisiae*. uORF activities are moderately well correlated, suggesting relaxed selection on the magnitude of repression. (**B**) Boxplot depicts the magnitudes of uORF activities for de novo uORFs (mutation of NCC-uORFs to AUG-uORFs), conserved uORFs (Start codon PhastCons ≥ 0.5) and not-conserved uORFs (PhastCons <0.5). Natural uORFs are much less repressive than de novo uORFs, suggesting strong selective pressure to remove novel uORFs. Conserved uORFs are also significantly less repressive than non-conserved uORFs. (**C**) Conserved uORF start codons are farther from transcription start sites than non-conserved uORF starts. (**D**) Kozak context expression scores from de novo, non-conserved, and conserved uORFs. Natural uORFs have a lower Kozak score than de novo uORFs, and conserved uORFs have a lower score than non-conserved, which likely contributes to their different magnitudes of repression. * indicates P < 0.05, and *** indicates P < 1 x 10^-10.

The online version of this article includes the following figure supplement(s) for figure 6:

**Figure supplement 1.** 6Contribution of Kozak context differences to divergence in the AIM22 uORF repression.

at 25 nucleotides upstream, and only 1.5-fold repressive when transcription is initiated a further 45 nucleotides upstream (*Figure 6B*). Both YFP mRNA and protein levels are extremely low in the former, suggesting that initiation at the downstream transcription start site is largely unproductive due to NMD and translational repression. A homologous uORF located in the *S. cereviaie NHEJ1* has similar dependence on transcription start site location, however its reporter expression values were too noisy for confident estimation. Similarly, a uORF in *S. cerevisiae DAL7* is not functional in the context of the longest (142 nt) transcript leader tested, but becomes a 1.3-fold repressor in an 124 nt leader and a 2.3-fold repressor when located just 11 nucleotides downstream of the 5' cap. However, there are exceptions to this general rule, as shown by a highly conserved uORF found upstream of *S. cerevisiae RNT1*. The *RNT1* uORF is a 1.3-fold repressor in its longest transcript leader context (93 nt), but shows weaker repression in two shorter transcript leader isoforms. Together, these results show a striking dependency of uORF activity on the location of transcription initiation, underscoring the importance of position in uORF function.

## Evolutionary constraints on the magnitude of uORF activity

While previous studies have evaluated uORF conservation, these have generally not investigated conservation of uORF activity. Thus, we examined the activities of orthologous uORFs from *S. cerevisiae* and *S. paradoxus*, which last shared a common ancestor ~5 Mya (*Scannell et al., 2011*). Notably, our data compare these uORFs in their native homologous transcript leaders. After removing uORFs whose activity measurements were noisy (σ>0.15) the direction of uORF regulation (repressor or enhancer) was almost entirely conserved (77/78; *Supplementary file 1h*). Conservation of the magnitude of regulation was generally well correlated ($R^2$=0.62, *Figure 6A*), with some exceptions. For example, an *S. cerevisiae* oORF in *SEC1* was 1.5–2-times more repressive than its *S. paradoxus* homolog. Notably, there is a deletion in *S. paradoxus* that results in an earlier stop codon that shortens the oORF. In another case, an *S. paradoxus* oORF in the *AIM22* leader was approximately fourfold more repressive than its *S. cerevisiae* homolog, possibly due in part to the presence of more adenosines in its Kozak sequence (*Figure 6—figure supplement 1*). In contrast, the sequence and location features of the *ECM7* uORF were highly conserved, and both species uORFs had strong 10-fold repression. Thus, the

magnitude of repression at many yeast uORFs has diverged substantially, though their functions as repressors are generally conserved.

We next divided uORFs by the average PhastCons Score (PCS) over their start codons into 'conserved' (PCS >0.5) and 'non-conserved' (PCS <0.5) groups. Phastcons scores range from 0 to 1.0, and represent the probability that a region has undergone negative selection (*Siepel et al., 2005*). Consistent with predictions from vertebrate Ribo-seq studies (*Chew et al., 2016*; *Johnstone et al., 2016*) conserved uORFs were less repressive than non-conserved (*Figure 6B*). We also found that mutating non-AUG uORF start codons to AUG created much more repressive 'de novo' uORFs (*Figure 6B*; *Supplementary file 1i*). These results suggest that newly emergent yeast uORFs tend to be strong repressors that are likely removed by natural selection. To investigate potential explanations for the decreased repression exhibited by conserved uORFs, we compared the features of 'conserved' and 'non-conserved' uORFs selected by our ENR modeling. Interestingly, conserved uORF start codons were located further from transcription start sites (*Figure 6C*) and had weaker Kozak contexts (*Figure 6D*, WRT p=0.0547) than their non-conserved counterparts. These results suggest that both sequence and position features are subject to selection, such that conserved uORFs tend to have features associated with milder impacts on expression.

## Discussion

Thousands of AUG and non-AUG uORFs have been identified from ribosome profiling studies in all major model organisms (*Kearse and Wilusz, 2017*). Despite concerns that cycloheximide and other drugs used in ribosome profiling may exaggerate occupancy on transcript leaders (*Gerash-chenko and Gladyshev, 2014*; *Kearse et al., 2019*; *Lareau et al., 2014*), very few natural uORFs have been experimentally tested because traditional uORF assays are performed on a gene-by-gene basis (*Wethmar et al., 2014*). Here, we used two Massively Parallel Reporter Assay systems, FACS-uORF and PoLib-seq, to quantify the impact of thousands of yeast AUG- and non-AUG uORFs on reporter expression and translation efficiency in wildtype and NMD-deficient yeast strains. We leveraged the resulting data to investigate the importance of multiple uORF features on their regulatory functions. Our analyses shed new light on uORF functions and have multiple implications.

It has long been recognized that eukaryotic translation can initiate at non-AUG (so-called) near-cognate codons (*Kearse and Wilusz, 2017*; *Kozak, 1991a*; *Tang et al., 2004*). Ribosome profiling studies also support the translation of many non-AUG uORFs in yeast (*Brar et al., 2011*; *Eisenberg et al., 2020*; *Ingolia et al., 2009*; *Spealman et al., 2018*). However, the few non-AUG uORFs that have been experimentally tested had relatively modest regulatory effects in reporter systems (*Spealman et al., 2018*; *Zhang and Hinnebusch, 2011*). Our results show non-AUG uORFs have relatively mild impacts on gene expression. Furthermore, a considerable number of non-AUG uORFs whose mutation significantly changed reporter expression may not be translated, as insertion of the stalling CGA dicodon sequence did not reduce expression. Thus, at least during log-phase growth, many non-AUG uORFs appear to have little influence on gene expression, consistent with the relatively low rate of translation initiation at near-cognate start codons (*Kolitz et al., 2009*; *Takacs et al., 2011*). As non-AUG uORFs have been predicted in many other species, our results suggest that most of them may also play limited roles in regulating expression. However, non-AUG uORFs with start codons in strong Kozak contexts (*Diaz de Arce et al., 2017*) with accompanying downstream mRNA structure (*Kozak, 1990*) may more have more substantial effects.

Although most uORFs are expected to repress expression, examples of enhancer uORFs have been identified in multiple species. Most notably, the short upstream uORFs in *GCN4* (yeast) and *ATF4* (metazoans) have been shown to cause stress-dependent upregulation of main ORF translation (*Hinnebusch, 2005*; *Vattem and Wek, 2004*). This upregulation results from delayed reinitiation under stress, such that PICs that resume scanning bypass more repressive uORFs. In our system, only two percent of uORFs increased expression. Notably, the oligonucleotide synthesis technology used to generate our reporter library is limited in length, such that the longest leaders that we tested were 180 nucleotides. While this included most yeast transcript leaders, longer leaders may have more enhancers. As such, our numbers may underestimate the frequency of enhancers. Because most of the enhancers we found were in multi-uORF transcript leaders, we propose that, much like *GCN4* uORF1, they function by reducing usage of other, more repressive uORFs. Although we performed our experiments under unstressed conditions, reinitiation might still allow leaky scanning past more repressive

uORFs near the enhancer stop codon. In other cases, translation of enhancer uORFs might alter transcript leader structure or modulate scanning by upstream PICs. Regardless of the mechanisms used, our data indicate that uORFs rarely act as enhancers, at least under the conditions we assayed.

It is well known that uORF activity depends on the strength of start codon recognition. However, the influence of other factors has been less clear. Our computational modeling of a massive set of yeast uORFs compared the predictive power of these and other features on uORF activity. While the correlation of Kozak context and uORF activity was strong, surprisingly, uORF location had similar predictive power. Because PIC assembly involves unwinding the mRNA around the 5′cap, the increased repression we observed for uORFs that start near the cap might result from increased start codon accessibility. Cap-proximal uORFs might also sterically block the loading of additional PICs. Previous work in yeast found that cap-proximal start codons were often skipped (*Arribere and Gilbert, 2013*); however, kozak context and RNA structure were not considered. Indeed, a similar tendency to skip cap-proximal start codons in wheat germ extracts could be overcome by the presence of an RNA stemloop downstream (*Kozak, 1991b*). We also observed stronger repression for longer uORFs and uORFs that terminate closer to the main ORF start codon. Such uORFs may permit less reinitiation at the main ORF, as ribosomes lose initiation factors after translating PICs that resume scanning would have less time to reacquire ternary complex before encountering the main ORF start codon. If so, our results suggest translation reinitiation is more common in yeast than currently appreciated (*Gunišová et al., 2017*). Future work is needed to evaluate these and other potential mechanisms underlying the importance of uORF location.

Despite the insights gained by our ENR modeling, a large amount of the variance among uORFs cannot currently be explained. This suggests that other features may have important influences on uORF activity. For example, previous work found that structural accessibility of uORF start codons, as measured by SHAPE probing, led to accurate estimates of uORF activity for the human antitrypsin-alpha gene (*Corley et al., 2017*). Based on this work, it was proposed that ribosomes often shunt past uORFs whose start codons are occluded in stable RNA structures (*Mustoe et al., 2018*). Although intriguing, this model seems to negate the role of helicases (e.g. DED1 and eIF4A) which unwind such structures during PIC scanning (*Guenther et al., 2018*; *Sen et al., 2019*; *Sen et al., 2015*; *Sharma and Jankowsky, 2014*). While we included predicted energies of unwinding around the start codon, our purely computational structural predictions contributed only minimally to our models. Given the large amount of uncertainty surrounding computational estimates of RNA structure, we expect that detailed measurements of structural accessibility would be beneficial to future modeling.

By comparing the magnitude of uORF repression in wildtype and *upf1Δ* yeast, we found the contribution of NMD to uORF repression (%NMD) ranges from 0 to 100%, with a median of 35%. Our work also provides insight into how uORF features impact their ability to induce NMD. Reinitiation after uORF translation can protect human mRNA from NMD (*Lindeboom et al., 2016*; *Zhang and Maquat, 1997*), and short uORFs are known to reinitiate more efficiently than long ones (*Gunišová et al., 2017*, *Gunišová et al., 2016*) likely due to increased retention of initiation factors (*Bohlen et al., 2020*; *Wagner et al., 2020*). Consistent with this, we found short uORFs induce less NMD than long uORFs. We also found NMD induction was lowest for uORFs terminating in UGA<u>C</u>. Since this stop codon context allows high rates of readthrough, it is possible such readthrough protects mRNA from uORF-induced NMD consistent with previous work (*Keeling et al., 2004*). Notably, AU-rich elements downstream of stop codons were associated with stronger uORF repression in wildtype, but not *upf1Δ* yeast. Such sequences may function as NMD-inducing *cis*-elements formerly thought to promote NMD by binding Hrp1p (*Culbertson and Leeds, 2003*; *Czaplinski et al., 1999*; *González et al., 2000*; *Peltz et al., 1993*; *Ruiz-Echevarría et al., 1998*). Thus, our work supports significant roles for many sequence features in determining uORF NMD induction.

We also investigated the relationship between uORF location in transcript leaders and NMD induction. Previous work using synthetic NMD targets found that NMD was more efficient when stop codons were located closer to the 5′ end of the *PGK1* coding sequence (*Cao and Parker, 2003*). We found the opposite relationship among uORF stop codons, such that stop codons were less likely to induce decay the closer they were located to the 5′ cap. Although these results at first appear discordant, this may reflect much larger distances between the cap and stop codon in this study (>66, 426, 675, and 957 nucleotides) than found in our library (0–150 nt). It is also possible that the effect of stop location we observed may instead reflect uORF length, as longer uORFs terminate, on average, farther from

the transcription start site than shorter ones. Thus, while our work suggests uORF position may also be an important feature impacting the efficiency of NMD, other features may underly this correlation.

Eukaryotic transcription often initiates at alternative sites, even in the relatively simple yeast (*Lu and Lin, 2019*; *Pelechano et al., 2013*; *Zhang and Dietrich, 2005b*). Such alternative transcription start sites alter the translation efficiency of their downstream genes (*Arribere and Gilbert, 2013*; *Rojas-Duran and Gilbert, 2012*; *Zydowicz-Machtel et al., 2018*). However, the extent to which uORF function depends on transcription start site usage has not been previously evaluated to our knowledge. Our results indicate that uORF activity often varies with alternative transcription start site usage. The dependence of uORF activity on transcript start sites may impart different translation efficiency and turnover rates for alternative transcript isoforms. This was broadly consistent with our modeling data, in that uORFs were more repressive when present in transcript leaders that initiated closer to the uORF start codon. If uORF start codons near the 5' cap are more structurally accessible, uORFs that initiate further downstream may be bypassed by ribosomal shunting, as occurs in the Cauliflower Mosaic Virus uORF and has been proposed recently to be more common (*Corley et al., 2017*; *Mustoe et al., 2018*; *Ryabova and Hohn, 2000*). Alternatively, PIC scanning may become more processive over time, such that more distal start codons are more easily skipped. Regardless of the underlying mechanisms, the dependence of uORF activity on location implies that caution must be taken when studying uORFs using ribosome profiling data because current methods do not connect uORF occupancy with specific transcript isoforms.

Finally, our study included the first direct comparisons of gene regulation from homologous uORFs from two species in their corresponding, native transcript leaders. We found the magnitude of uORF regulation varied substantially for many uORFs, with some examples consistent with changes in uORF features. Our results also show a conserved uORF in yeast *ECM7* confers strong repression in both *S. cerevisiae* and *S. paradoxus*. Although previous work reported the uORF had no impact on regulation (*Zhang and Dietrich, 2005a*), this may have been due to non-physiological transcription initiation site usage. The luciferase reporter used in the previous study was driven by a different promoter which may have initiated transcription at alternative site(s). Earlier studies comparing ribosome profiling data from zebrafish, mouse, and human cell lines indicated that conserved uORFs tended to have weaker Kozak sequences than non-conserved uORFs (*Chew et al., 2016*; *Johnstone et al., 2016*). They proposed that this allowed conserved uORFs to be activated by *trans*-acting factors. In addition, conserved uORFs were associated with smaller decreases in ribosome occupancy on downstream main ORFs. By directly assaying uORF effects, we found that conserved uORFs indeed have more modest effects on gene expression than non-conserved uORFs. We also found uORFs with conserved start codons had weaker Kozak sequences and were located further from the transcription start site. These results suggest that both uORF Kozak context and location are subject to purifying selection, underscoring the importance of uORF position we identified by computational modeling. Future work could more fully investigate the evolution of yeast uORFs by evaluating the frequency of uORF-related mutations in the 1000 *S. cerevisiae* genome project data (*Peter et al., 2018*). However, such analysis would carry the caveat that the locations of transcript leaders in these strains have not been experimentally determined.

In providing the first, to our knowledge, high-throughput functional analysis of natural uORF regulatory activities, our work defines the range of uORF activity and reveals location is as important as Kozak strength in determining yeast uORF functions. Recent work has also highlighted the large number of human uORFs (*Barbosa et al., 2013*; *Calvo et al., 2009*; *Lee et al., 2020*; *McGillivray et al., 2018*; *Wethmar et al., 2010*). While many aspects of translation initiation are deeply conserved, the mechanisms involved in transcript leader scanning may differ substantially in human cells and tissues. Thus, studies using similar methods in human cells are needed to characterize human uORF regulatory functions, and further understand their potential involvement in cellular stress responses.

## Methods

### FACS-uORF library construction

The reporter plasmid was modified from *pGM-YFP-mcherry* (*Lin et al., 2019*) by replacing the GPMI promotor with an *ENO2* promotor. The *ENO2* promotor transcription start site is highly stringent (*Spealman et al., 2018*), allowing for highly consistent transcription starts sites in the transcript

leader library (*Figure 1—figure supplement 2*). The *ENO2* promotor was inserted upstream of YFP by amplifying the ENO2 sequence from *S. cerevisiae* genomic DNA, using the primers Eno2_SalI-F and Eno2_AvrII_R1. The reverse primer introduced an AvrII restriction site in *ENO2* for use during subsequent cloning steps. The PCR product was then amplified using the primers Eno2_SalI-F and Eno2_XmaI_R2 to add additional Eno2 sequence and an XmaI site. The ENO2 PCR product was digested with SalI and XmaI, and ligated into the vector, *pGM-YFP-mCherry* resulting in the plasmid construct *pGM-ENO2-YFP-mCherry*.

The uORF library was synthesized as two pools of 130- and 210-mer oligonucleotides (Agilent Technologies). Similarly, a Kozak-context library was synthesized using degenerate bases at the –4 to +1 position around the start codon sequence (IDT). Each oligonucleotide in the pool was designed to have a common *ENO2* promotor sequence and AvrII site upstream, and a yellow fluorescent protein (YFP) sequence downstream of the transcript leader so that the pool of oligos could be amplified and cloned into a dual fluorescent reporter. The oligo pool was provided as a 10 pmol pellet and was dissolved in 100 μl of TE. The primers Eno2_lib_F1and FACS-uORF-YFP-R, were used to amplify 30 μl (3 pmol) of the oligo pool in a 400 μl reaction (split up in 50 μl aliquots) containing 1 X Herculase II reaction buffer, 0.4 mM each dNTP, 0.25 μM each primer, and 16 μl Herculase II Fusion DNA Polymerase (Agilent Technologies). The reaction conditions for the PCR reaction were 95 °C for 1 min, followed by 10 cycles of 95 °C for 20 s, 55 °C for 20 s and 68 °C for 20 s, and then one cycle of 68 °C for 4 min. The PCR product was purified using AMPure XP beads (Agencourt) and resuspended in 20 μl of water. Half of the PCR product (9 μl) was used as a template for a second round of amplification, using the primers Eno2_lib_AvrII_F2 and FACS-uORF-YFP-R in a 400 μl reaction containing 1 X Q5 polymerase reaction mix, 0.2 mM each dNTP, 0.5 μM each primer, and 8 units of Q5 High-Fidelity DNA polymerase. The amplification conditions were 98 °C for 30 s, followed by 10 cycles of 98 °C for 10 s, 55 °C for 20 s and 72 °C for 30 s, and then one cycle of 72 °C for 2 min. The PCR product was purified over a Zymo Clean and Concentrator column (Zymo Research), digested with AvrII and BglII, and ligated into *pGM-ENO2-YFP-mcherry*. The ligated plasmids were then transformed into competent *E. coli*. Positive transformants were selected on LB plates containing ampicillin. In order to maintain the complexity of the library, a total of 100,000 positive colonies were scraped off 45 10 cm plates and pelleted. The plasmid library was extracted using a QIAGEN plasmid maxiprep column (Qiagen) and resuspended in 1 ml of TE.

## Yeast transformation

Competent cells for each yeast strain were prepared using the Frozen-EZ Yeast Transformation II Kit (Zymo Research) according to the manufacturers' instructions. For each strain, 400 μl of competent cells were mixed with 2 μg of each plasmid library and incubated shaking at 30 °C for 2 hr. To test the transformation efficiency, 10 μl of cells were plated on minimal media plates lacking uracil and incubated for 48 hr at 30 °C. The remaining cells were added to 30 ml of URA- media and incubated overnight shaking at 30 °C. The next day, the cells were added to 200 ml of synthetic complete media lacking uracil and incubated overnight shaking at 30 °C. To ensure that at least 100,000 individual cells were transformed, the number of colonies from 10 μl of transformed cells was counted. For all strains, the total number of total clonal transformants ranged from 500,000 to 1 million. Glycerol stocks were made from each uORF library in each strain, by pelting cells from 10 ml of overnight cultures.

## Fluorescence activated cell sorting (FACS)

For each strain of yeast, one glycerol stock tube (10 ml of overnight culture) of each uORF library was culture was added to 200 ml of synthetic complete media lacking uracil and grown shaking overnight at 30 °C. The cells were then restarted in 50 ml of URA- media at an $OD_{600}$ of 0.1–0.2 and grown shaking 30 °C to an $OD_{600}$ of 0.7, prior to FACS. Just prior to cell sorting, 12 ml of cells were pelleted and flash frozen for later RNA extraction, and 1 ml of cells were pelted and frozen for DNA extraction. The cells were sorted into nine bins, based on the ratio of YFP to mCherry, on a FACSVantage Digital Cell Sorter. For each bin, 100,000 cells were deposited into culture tubes containing 5 ml of URA-glucose media, and grown overnight shaking at 30 °C. To verify that the cells were sorted into the correct bins, and to later adjust the bin values to the ratio of YFP and mCherry, the ratio of YFP over mCherry for each bin was measured on a Tecan M1000 plate reader.

## Sequencing Library preparation

For the RNA sequencing libraries, 5 µg of total RNA was subjected to Dnase treatment, re-extracted with acid phenol, ethanol precipitated over a column, and resuspended in nuclease free water. Two µg of Dnase treated total RNA was reverse transcribed with a RT primer (FacsuORF-RT) that anneals to the YFP sequence downstream of subsequent PCR primers. The cDNA was ethanol precipitated and resuspended in 10 µl of nuclease free water. The adapter, 'Fuorf_RNA_adapter' was ligated to the 3' end of 10 µl of cDNA in a 20 µl reaction at 65 °C for 1 hr using Thermostable 5' App DNA/RNA ligase (New England Biolabs). The reaction was inactivated at 95 °C for 3 min and 2 µl of the reaction was used as a template for PCR for sequencing library generation as described below.

The plasmid DNA libraries from the unsorted library and from each bin were prepared by three rounds of PCR. Using primers that anneal to the ENO2 promoter (Eno2_lib_F1), and YFP sequences (FACs-uORF-YFP-R) 8 cycles of PCR were performed in a 20 µl reaction using 10 ng of plasmid library (or 2 µl of cDNA, as described above) as a template. The first PCR reaction was purified using 1.5 X AMPure XP beads (Agencourt) and resuspended in 10 µl of water and added to a second reaction where 6 cycles of PCR was performed using primers that included varying numbers of random bases to stagger the sequencing reads (FuORF_2_DNA_N0-N7_F, and FuORF_2_DNA_N0,2,6_R). The second PCR was purified using 1.5 X AMPure XP beads (Agencourt) and resuspended in 50 µl of nuclease free water. Using 2 µl of the second PCR as a template, a third round of 15 PCR cycles was used to incorporate Illumina sequences and 6 nucleotide barcodes, using the primers RPF_F and RPF_R. All PCR amplifications were performed using a high-fidelity polymerase (Q5 DNA polymerase, New England Biolabs) to avoid PCR introduced sequence error.

## PoLib-seq

One glycerol stock tube (10 ml of overnight culture) of each uORF library was added to 200 ml of URA-media and grown shaking overnight at 30 °C. The cells were then restarted in 50 ml of URA- media at an $OD_{600}$ of 0.2 and grown shaking 30 °C to an $OD_{600}$ of 0.7 and harvested by vacuum filtration. The cells were scraped off of the filter and flash frozen in liquid nitrogen. The yeast were cryoground using a mortar and pestle in 1 ml frozen polysome lysis buffer (PLB, 10 mM Tris-HCl pH 7.5, 0.1 M NaCl, 30 mM $MgCl_2$, 50 mg/ml heparin, 50 mg/ml cycloheximide). The ground frozen yeast were added to 1.5 ml fresh PLB in a 2 ml microcentrifuge tube and thawed on ice. Approximately 0.5 ml of 0.5 mm zirconia/silica disruption beads were added to the lysates and the lysates were ground by vortexing for 30 s and then cooled for 30 s on ice for a total of four grinding cycles. The lysates were cleared by centrifuging for 10 min at 20,000 G at 4 °C. The $OD_{254}$/ml were determined and the lysates were flash frozen in liquid nitrogen in 50 $OD_{260}$ aliquots. Forty $OD_{260}$ units of lysates were layered on 7–47% (w/v) sucrose gradients, centrifuged (4 hr, 4 °C at 27, 000 rpm) using Beckman L7 ultracentrifuge. A Teledyne ISCO Foxy R1 density gradient fractionator was used to fractionate and analyze gradients with continuous monitoring at $OD_{254}$. Ribosomal subunits, ribosomes, and polyribosomes were fractionated according to the manufacturer's protocol, and appropriate fractions ('top fraction', 40 S, 60 S, monosome, 2, 3, 4, and greater than 5 ribosomes) were collected. RNA from each fraction ('top fraction', 40 S, 60 S, monosome, 2, 3, 4, and greater than 5 ribosomes) was extracted by two rounds of acid-phenol extraction, purified over RNA clean up and concentrator – 5 (Zymo Research) columns, and eluted in nuclease free water. RNA sequencing libraries were prepared from each fraction (see Sequencing library preparation).

## FACS-uORF and PoLib-seq sequencing and data analysis

FACS-uORF and PoLib-seq libraries were subjected to paired-end (2x150 cycle) Illumina sequencing (Novogene). Read pair data were merged and error corrected using FLASH (*Magoč and Salzberg, 2011*) using parameters '-z -O -t 1 M 150'. ENO2 promoter plasmid sequence present in DNA libraries (FACS only) was removed from the resulting merged reads using cutadapt (*Martin, 2011*) with parameters '--trimmed-only -e 0.04 g AGTTTCTTTCATAACACCAAGC' . The resulting trimmed reads from each FACS bin were separately counted for perfect matches to the designed library UTR sequences using a custom perl script (DNA-seqcount.pl). YFP reporter RNA library data were merged and error corrected using FLASH as above. The resulting merged RNA-seq reads were further processed with cutadapt parameters '--trimmed-only -e 0.04 g AGTTTCTTTCATAACACCAAGCNNNN' to remove the 3' Fuorf_RNA_adapter sequence ligated to cDNA during RNA library preparations. After this adapter

trimming step, many resulting RNA-seq reads contained two random bases and a guanosine, 'NNG', preceding the designed transcription start site sequence 'AAGC' from the ENO2 transcript leader. On their 5' ends. These 'extra' sequences likely reflect reverse transcription of the 5' 7 mG cap (the universal "G") followed by the untemplated addition of two random bases during cDNA synthesis. Cutadapt was used to remove these sequences, with parameters '--trimmed-only -g ^NNGAAGC' to identify 5'-capped reads and '--discard -g ^NNGAAGC' to identify uncapped reads. The resulting reads from each sample were counted for perfect matches to the designed UTR library using the custom perl script (RNA-seqcount.pl).

## FACS-uORF analysis

For each of the three FACS-uORF replicates, count data from the RNA-seq library, total plasmid library ('bin0'), and libraries from each of the eight FACS bins (bin1 - bin9) were normalized and processed as previously described (*Lin et al., 2019*) using a custom perl script ('FuORF-Tables-Maker. pl'). Briefly, the relative representation of DNA and RNA reads was calculated as the reads-per-million for each replicate, and relative transcript levels were estimated as RNA-rpm / DNA-rpm. YFP/mCherry expression levels for each bin were taken from TECAN readings of overnight sorted cultures (see FACS methods) and normalized to the highest value (bin8=1). Read count data from each bin were downsampled such that the proportion of total reads from each bin reflected the proportion of cells sorted into that bin. After these normalization steps, the average YFP value for each UTR in the library was calculated as follows: YFP / mCherry = (SUM(YFPbin * reads / bin) from bin 1–9/total reads). The resulting output files contained estimates of RNA and YFP levels for each UTR. The three replicates were compared, and UTRs with noisy YFP measurements (standard deviation >0.05, minimum of 50 normalized reads per UTR) were removed from further consideration (*Figure 1—figure supplement 1*). For each uORF, mean YFP values from wildtype and mutant UTRs compared using the WRT in each replicate, and the benjamini - hochberg correction was applied to control the FDR at 5%. uORFs with significant (FDR <0.05) and consistent (same direction) effects on YFP levels were considered significant regulators. Statistical analyses were done using R (version 3.6.1).

## PoLib-seq analysis

PoLib-seq libraries were pooled to provide sequence read counts in proportion to the total RNA present in each sucrose gradient fraction (40 S, 60 S, monosome, 2, 3, 4, 5+) (*Figure 2—figure supplement 2*). To compensate for variations in manual fractionation of the replicate sucrose gradients, we pooled 'translating' (2, 3, 4, and 5+ribosomes) and 'non-translating' (40 S, 60 S, and monosome) fractions separately for each replicate, similar to RNA-seq based polysome profiling analyses (Anota2seq, e.g. *Oertlin et al., 2017*). The 'top' fraction was not used for analysis. After manual inspection, the monosome fraction of replicate 1 was downsampled by a factor of 0.8626 to ensure similar proportions of 'translating' (45%) and 'non-translating' (55%) reads from each of the two replicates (*Figure 2—figure supplement 2*). A 5000 total read cutoff (capped and uncapped, replicates 1 and 2 summed across all fractions) was used for each wildtype / mutant uORF comparison. The impact of each uORF on ribosome loading was calculated as the fold change in the translated / untranslated ratio from the mutant UTR compared to the wildtype UTR using a custom perl script (PoLib-Tables-Maker.pl). uORFs with consistent (same direction) influences on expression were tested for significance using the cochran-mantel haenszel test, and an FDR of 5% was maintained after benjamini-hochberg correction of raw p-values. Statistical analyses were done using R (version 3.6.1).

## uORF feature compilation

Features were compiled for elastic net regression modeling (*Supplementary file 1j and k*) as follows. Kozak scores were taken from 8,096 reporters in which the sequence was varied around AUG and seven NCC YFP start codons (i.e. NNNNATGN, NNNNATTN, NNNNATAN, NNNNATCN, NNNNACGN, NNNNGTGN, NNNNTTGN, NNNNCTGN; *Figure 4—figure supplement 1*). Phastcons conservation scores (*Siepel et al., 2005*) were downloaded from the UCSC genome browser and the average conservation score was calculated over the nucleotides for each start codon and each stop codon, separately. These scores were only available for *S. cerevisiae* uORFs. For *S. paradoxus* uORFs, these scores were set to the average conservation rate of *S. cerevisiae* uORFs. Orthologous uORFs were defined by having start codons at the same position in two-way alignments between *S. cerevisiae* and

*S. paradoxus*, produced using EMBOSS Needle (*Madeira et al., 2022*). The energies of unwinding the RNA around uORF start codons (ΔΔG) were estimated based on previous work (*Corley et al., 2017*) using programs from ViennaRNA (*Lorenz et al., 2011*). RNAsubopt was used to generate one hundred suboptimal folded structures for each wildtype 5' UTR, including 50 nucleotides of YFP. uORF-start codon unfolded structures were generated by setting the –15 to +15 region around each uORF start codon as unpaired. RNAeval was used to calculate the ΔG of each folded and unfolded structure, which were subtracted to generate one hundred estimates of the ΔΔG of unfolding for each uORF, the medians of which were used as modeling features. The %AU was calculated for the +2 to+10 region downstream of uORF stop codons. uORF peptide charges (isoelectric points) were calculated using EMBOSS Pepstats (*Madeira et al., 2022*), as positively charged residues have been associated with ribosome stalling (*Requião et al., 2016*). The Proline / Glycine frequency was calculated as the number of proline and glycines divided by the peptide length because proline and glycine codons are enriched at sites of ribosome stalling (*Artieri and Fraser, 2014*; *Ingolia et al., 2011*).

## uORF modeling and feature selection by elastic net regression

Elastic net regression attempts to solve the feature-selection and shrinkage problems simultaneously with the introduction of both $L_1$ and $L_2$ regularization. The $L_1$ parameter functions similarly to LASSO regression and selects features, and the $L_2$ parameter minimizes the coefficients of predictors (*Hastie et al., 2009*). The model minimizes the objective function:

$$\frac{1}{2n} \|y - Xw\|_2^2 + \lambda_1 \|w\|_1 + \lambda_2 \|w\|_2^2$$

With and representing the $L_1$ and $L_2$ regularization parameters respectively. Regression was performed using the sklearn ElasticNet package, which takes two hyperparameters. The score penalty α corresponds to the magnitude of the regularization penalties, and the *l1_ratio* parameter () is used to control the relative values of the $L_1$ and $L_2$ penalties, such that:

$$\lambda_1 = \alpha\ell, \ \lambda_2 = .5\alpha(1 - \ell)$$

Fifteen features were selected from the data and were normalized by minmax. Test and training data were separated according to an 80/20 split. The hyperparameters α and were tuned manually to balance model performance and precision over 100 trials with randomly initialized training and test data. Model performance in this case refers to the R2 score of the model on the test set, and precision refers to the similarity of selected feature subsets between trials.

It has been previously shown that the strength of the Kozak sequence is strongly implicated in the translational regulation of uORFs. For this reason, single-feature linear regression on Kozak strength alone was used as a control to demonstrate the robustness of the model. For each of the 100 trials, the R2 score of the Elastic Net model against single-feature LR was determined, and p-values were computed using the WRT.

## Acknowledgements

This work was supported by the National Institute of General Medical Sciences (grant R01GM121895 to CJM and R01GM028301 to JW). This work was supported by the National Institute of General Medical Sciences to CJM (grant R01GM121895) and to JW (grant R01GM028301). The authors would like to thank Dr. Andreas Pfenning for advice on elastic net regression modeling.

## Additional information

### Funding

| Funder | Grant reference number | Author |
|---|---|---|
| National Institutes of Health | R01GM121895 | Gemma E May<br>Matthew Agar-Johnson<br>Joel McManus<br>Christina Akirtava |

| Funder | Grant reference number | Author |
|---|---|---|
| National Institutes of Health | R35GM145317 | Gemma E May<br>Christina Akirtava<br>Joel McManus |
| National Institutes of Health | R01GM028301 | Jelena Micic<br>John Woolford |

The funders had no role in study design, data collection and interpretation, or the decision to submit the work for publication.

### Author contributions
Gemma E May, Formal analysis, Investigation, Visualization, Methodology, Project administration, Writing – review and editing; Christina Akirtava, Data curation, Software, Formal analysis; Matthew Agar-Johnson, Data curation, Software, Formal analysis, Visualization, Methodology; Jelena Micic, Investigation, Methodology, Writing – review and editing; John Woolford, Supervision, Funding acquisition, Writing – review and editing; Joel McManus, Conceptualization, Data curation, Software, Formal analysis, Supervision, Funding acquisition, Visualization, Writing – original draft, Project administration, Writing – review and editing

### Author ORCIDs
Joel McManus (iD) http://orcid.org/0000-0002-6605-2642

### Decision letter and Author response
Decision letter https://doi.org/10.7554/eLife.69611.sa1
Author response https://doi.org/10.7554/eLife.69611.sa2

# Additional files

### Supplementary files
• Supplementary file 1. Data tables. This file contains data tables that have (a) expression differences between 2,033 wildtype uORF transcript leader reporters and corresponding AAG start codon mutations in the longest 5' UTR assayed for each uORF, (b) a list of uORFs and host UTRs that whose mutation decreases expression but are likely false-positives, (c) Expression differences between 417 wildtype uORF transcript leader reporters and corresponding CGACGA insert mutations, (d) expression differences between 2,209 wildtype uORF transcript leader reporters and corresponding AAG start codon mutations using PoLib-seq (minimum 5000 reads), (e) expression differences between 1,360 wildtype uORF transcript leader reporters and corresponding AAG start codon mutations in the longest 5' UTR assayed for each uORF, (f) expression levels (YFP / mCherry) for 8,096 Kozak variants of AUG and NCC start codons, (g) uORF activities in alternative transcript leaders, (h) Activities of homologous uORFs from *S. cerevisiae* and *S. paradoxus*, (i) expression differences between 285 wildtype uORF transcript leader reporters and corresponding AUG start codon mutations, (j) features used for elastic net regression modeling of uORF activities in the wildtype strain, and (k) features used for elastic net regression modeling of uORF activities in the UPF1 deletion strain.

• Transparent reporting form

### Data availability
Sequencing data have been deposited in NCBI SRA under accession PRJNA721222.

The following dataset was generated:

| Author(s) | Year | Dataset title | Dataset URL | Database and Identifier |
|---|---|---|---|---|
| May GE, Akirtava C, Matthew Agar-Johnson M, Micic J, Woolford J, McManus J | 2021 | *Saccharomyces cerevisiae* translation massively parallel reporter assay raw reads | https://www.ncbi.nlm.nih.gov/bioproject/PRJNA721222 | NCBI BioProject, PRJNA721222 |

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
