## [Editor Report]

This important study advances our understanding of how upstream reading frames contribute to gene expression regulation. Using innovative tools, the authors provide convincing evidence connecting the features of these sequences to protein expression. The results will be of broad interest to investigators in the field of gene expression regulation and its evolution.

---

## [Decision Letter]

**Decision letter after peer review:**

Thank you for submitting your article "Unraveling the influences of sequence and position on yeast uORF activity using massively parallel reporter systems and machine learning" for consideration by *eLife*. Your article has been reviewed by 2 peer reviewers, one of whom is a member of our Board of Reviewing Editors, and the evaluation has been overseen by Naama Barkai as the Senior Editor. The reviewers have opted to remain anonymous.

Essential revisions:

The essential revisions are outlined in the referees' reports and summarized here:

(1) Estimate the extent to which the effects measured are independent of the reporter used.

(2) Strengthen the analysis of evolution of uORFs between *S. cerevisiae* and S. paradoxus, for instance by testing their interpretations.

(3) Show that the reporter mRNA initiates at the predicted TSS.

(4) Demonstrate experimentally that the distance from the cap causes the effects revealed from the statistical analyses.

*Reviewer #1 (Recommendations for the authors):*

This is a very impressive set of experiments and a timely paper.

I have two main comments. One is already explained in the public review and regards the potential dependency of the effects measured on the main ORF. I think this could be discussed but it could be also rather simple to test a subset of uORFs across the range of effects, from strongly repressive to strongly enhancing, and use a different reporter sequence to show that at least the ranking or sign of effects is preserved. Could be done by swapping mCherry and YFP for instance in the plasmid.

This is probably because of my background but I found the section on evolution really interesting and broadening the scope of the paper quite a lot. I wonder if the authors could not push further their analysis for instance by analyzing the polymorphism data available for 100s of strains in *S. cerevisiae*, for instance by looking at the frequency spectrum of mutations that would enhance the repressive effects of uORFs. Mutational bias could also be considered. With this part, it would also be interesting to know whether there are interactions between uORFs and the main ORFs because changes in magnitude or even signs of uORFs effects could be compensated by other regulatory changes, in a coevolution akin to what the same authors have reported on before regarding transcription and translation.

*Reviewer #2 (Recommendations for the authors):*

– They have not verified that a random sampling of reporter mRNAs generally have 5' ends corresponding to those predicted by using the ENO2 promoter in their library design. This is important because the position of uORFs from the 5' end is claimed to be an important determinant of uORF function, and represents one of the most novel findings of the study.

– They have not tested their conclusions about the importance of distance of the uORF from the mRNA cap on uORF function by follow-up experiments on a few individual repressive uORFs to examine whether increasing the distance of the uORF ATG from the cap by insertion of unstructured sequences (eg. CAA repeats) reduces inhibition by the uORF; or whether deleting sequences to move a uORF closer to the 5' end increases its inhibitory effect. This is important because the conclusion is based only on correlation analysis and hence could be influenced by other features of the mRNA that vary in parallel with distance of the uORF from the cap.

– Regarding the distance from the cap effect, it seems possible that repression by an uORF would be diminished by a location further downstream from the cap owing to inclusion of other AUG or near-cognate start sites located upstream of the uORF, which would reduce translation initiation at the uORF of interest. This possibility should be addressed bioinformatically, and also by examining individual reporters in which all AUG or near-cognate triplets are removed upstream of the uORF in question to determine if this increases repression by the uORF without changing its position relative to the cap.

– A point of major confusion: Most of the plots and supplementary tables list the reporter expression ratios as log2(AAG/WT); however, I believe that this incorrect and the log2(WT/AAG) ratios have been plotted instead, as the values for the majority of uORFs, which are repressive, are <0.

­– p. 13: The sentence: "In general, repressor uORFs were less repressive in the upf1∆ strain, including 92% of AUG-uORFs and 59% of non-AUG uORFs." requires statistical analysis to bolster the claims, particularly for the small subset of repressive non-AUG uORFs

– p.14: The sentence "Consistent with this, we found median %NMD was higher for uORFs terminating with UGA (37.5%) than those terminating with UAA (35.3%) or UAG (33.2%) (Figure 3D)." is not justified, as the median values for the three stop codons in Figure 3D do not appear to differ significantly. Moreover, assuming it was valid, is there any way to rationalize it based on known termination efficiencies at the three stop codons?

– Regarding the claim on p. 14 that "Unexpectedly, the location of the stop codon relative to the transcript leader cap was positively correlated with the %NMD, such that uORFs that terminate further from the cap were more likely to induce NMD than those that terminate adjacent to the cap (Figure 3F; R^2 = 0.065; P = 7.44x10^-8)", it should be noted that this is a very weak trend that explains only ~7% of the variance in %NMD values. As such, the sentence in the Discussion "We found the opposite relationship among uORF stop codons, such that stop codons were less likely to induce decay the closer they were located to the 5' cap" appears to be an overstatement. Also, isn't it possible that the trend can be explained as originating from uORF length effects shown in panel F, as longer uORFs will have greater cap to stop codon separations? (Note also that panels F and G were incorrectly cited in text.)

– pp. 16-17 and Figure 4D: The description of the parameters included in the ENR model is wholly inadequate. How was Kozak context quantified and for what sequence interval around the AUG? How was uORF Start and Stop and CDS sequence conservation quantified (and for what species)? What sequence interval around the uORF Start was employed to calculate the deltaG folding energies, and how were the calculations made? How far downstream was %AU calculated. In general sufficient information has to provided in the legends or Methods to allow the analysis to be repeated in full by other workers. A related comment is that the authors cite no literature to justify their analyses of these different parameters which they seem to pull out of the hat, eg. why Pro and Gly codons?

– p.19 and Figure 5A: the claim that "In general, uORFs were more repressive when they were closer to the TSS" is not convincing, as many uORFs shown in blue in Figure 5A show the opposite trend; and there are only a few outliers that conform to the stated trend. No statistical analysis of the trend was provided. As such, it is an overinterpretation of the data to claim that these results independently support the importance of uORF distance from the TSS, indicated by the ENR analysis. It also seems important to provide the YFP expression for the different leaders with WT and mutant uORFs, rather than just the WT/AGG ratios, in order to evaluate whether there are other sequences besides uORFs in the longer 5'UTRs that are affecting expression. This information should be added to Figure 5B in particular.

– p.20-21" Regarding the text: "For example, an *S. cerevisiae* oORF in SEC1 was 1.5 to 2-times more repressive than its S. paradoxus homolog, potentially owing to a deletion in S. paradoxus that results in an earlier stop codon that shortens the oORF. In another case, an S. paradoxus oORF in the AIM22 leader was approximately four-fold more repressive than its *S. cerevisiae* homolog, possibly due to the presence of more adenosines in its Kozak sequence." The words "potentially" and "possibly" in these sentences presumably reflect the fact that there are other sequence differences in the 5'UTRs between the two species that contribute to the differences in uORF function. These interpretations should be tested by mutational analysis of the reporters for these genes to determine whether shortening the oORF in S.c. SEC1 and improving the Kozak context of the S.c AIM22 oORF are sufficient to confer the altered repressive functions of the orthologous oORFs in S.p.

–

Figure 6D and related text: It is not convincing that conserved uORFs have a statistically significant poorer AUG context compared to nonconserved uORFs, even if one focuses (as they do) on only the -3 position rather than calculating the context score for the entire sequence interval surrounding the AUG. This is being overinterpreted.

---

## [Author Response]

Essential revisions:The essential revisions are outlined in the referees' reports and summarized here:(1) Estimate the extent to which the effects measured are independent of the reporter used.

To estimate the influence of the reporter protein on uORF activity measurements, we made reporters to test 6 uORFs with various effect sizes in which we swapped the fluorescent reporters, placing the uORF upstream of mCherry, while YFP became the internal control. We also recloned the same uORFs upstream of YFP, with mCherry as an internal control. We then tested all of the reporters using a TECAN fluorometer. This was done to ensure the experimental measurements were equivalent. Considering all 6 uORFs, the Pearson’s correlation constant (R) was 0.814 (R-squared = 0.662). However, one of the reporters had very low expression and a high variance by TECAN measurement. Excluding that reporter uORF, the R-square was 0.869 and R was 0.932. We conclude our results show that the reporter protein has minimal impact on the uORF effect size. These results are now included in the new Figure 1—figure supplement 3 and noted on page 8 of the revised manuscript.

(2) Strengthen the analysis of evolution of uORFs between *S. cerevisiae* and *S. paradoxus*, for instance by testing their interpretations.

To further evaluate the evolutionary differences between *S. cerevisiae* and *S. paradoxus* uORFs, we swapped the Kozak regions of the homologous uORF in gene YJL046W (AIM22) and compared their activity by TECAN fluorometer measurement. Consistent with our expectation, the *S. cerevisiae* Kozak region decreased repression in the *S. paradoxus* uORF. Similarly, the *S. paradoxus* Kozak sequence increased repression in the *S. cerevisiae* uORF. This result has been added to the manuscript in the new Figure 5—figure supplement 1. We also revised the analysis of evolutionary changes in uORF activity to compare numerical Kozak context scores for conserved, non-conserved, and de novo uORFs. These analyses supported our initial interpretations from the first submission.

(3) Show that the reporter mRNA initiates at the predicted TSS.

Transcription start sites can be identified by the presence of a 5’ 7-methyl-guanosine (m7G) cap immediately before the site of initiation. Notably, the reverse transcriptase we used in our assay is known to add a cytosine at the site of the m7G cap during cDNA synthesis, followed by two additional nucleotides, primarily CNN (Wulf et al., JBC 2019; Figure 3; PMID 31640989), where the underlined C is complementary to the m7G cap. Because we generated our RNA libraries by ligating an oligo to the 3’ end of the resulting cDNA, reads resulting from capped mRNA contain the complementary sequence “NNG” before the transcription start site, where the “G” reflects the m7G cap nucleotide. In essence, our RNA-seq approach is also a form of CAGE-seq. The transcript leaders in our library are designed to initiate with “AAGC” (from ENO2) immediately before each transcript leader sequence. Thus, “capped” RNA-seq reads will begin with “NNGAAGC” on their 5’ ends.

We reasoned we could use this property of our RNA-seq data to evaluate the frequency at which the capped mRNA initiated at their designed transcription start sites To identify capped RNA-seq reads, we first filtered out reads that aligned perfectly end-to-end to the library design. Then we selected all of the remaining reads that had an “NNG” at their 5’ ends, and trimmed off the NNG. These “cap possible” reads were then aligned to the designed library sequences. Reads that only matched the library design after trimming a 5’ NNG sequence were considered “capped” reads, such that their first base is the transcription start site. In each replicate, ~ 96.6% of the capped reads mapped precisely to the designed AAGC transcription start site. Thus, we find ~ 97% of transcripts from the reporter library initiate transcription at the designed location. We added a new supplemental figure (Figure 1—figure supplement 2) showing this analysis and updated the text accordingly (page 8 of the revision).

(4) Demonstrate experimentally that the distance from the cap causes the effects revealed from the statistical analyses.

Our model suggested uORFs that are far from the TSS will become more repressive when closer to the TSS, and vice versa. To test this, we extended the UTRs of *S. cerevisiae* YDR396W and *S. paradoxus* YHR039C, which have strong and medium repressor uORFs, by adding 72-nt of sequence to the UTR 5’ end. The added sequence was designed to prevent the formation of RNA structure using computational predictions (Reuter and Mathews, 2010). The YDR396W uORF became less repressive, while the repressiveness of the YHR039C uORF did not change. Reciprocally, we deleted the 5’ sequence of the YML007W gene’s 5’ UTR to place the uORF 10 nt from the 5’ cap. This did not change the repressiveness of the uORF. We also reduced the UTR length of YPR059C, a weak repressor, to place its uORF 10 nt from the 5’ cap. Contrary to our expectations, the uORF became less repressive. These results have been added to the new supplemental Figure 4—figure supplement 2 and discussed on page 17 of the revised manuscript.

While these results show that uORF repression varies with location in the 5’ UTR, they are not completely consistent with our regression model. One uORF changed as expected, two did not change and one changed in the opposite direction. This may reflect additional effects of changing the 5’ UTR that are not entirely related to uORF position. For example, though we attempted to design the UTR extensions to be unstructured, they may still alter the structural context of uORF or YFP start codons. Some of these sequences could also recruit RNA binding proteins. Similarly, our truncation mutations may have altered the structural accessibility of start codons or removed sites of *trans*-acting factors. We believe our regression modeling accounts for such additional factors, by leveraging results from thousands of uORFs in diverse sequence, structural, and location contexts. We also performed further computational analysis in response to a comment from reviewer #2, and found that the location effect was seen for single uORF transcript leaders, indicating it is not due to interactions among uORFs.

This highlights an important aspect of computational modeling in general – that correlations are not necessarily causative. Models are useful to the extent that they allow predictions of outcomes. We have revised our manuscript to make this more apparent, now explicitly noting that uORF locations are predictive of their function (revisions on pgs 2, 7, 17, and 25).

Reviewer #1 (Recommendations for the authors):This is a very impressive set of experiments and a timely paper.

We thank the reviewer for these encouraging comments

I have two main comments. One is already explained in the public review and regards the potential dependency of the effects measured on the main ORF. I think this could be discussed but it could be also rather simple to test a subset of uORFs across the range of effects, from strongly repressive to strongly enhancing, and use a different reporter sequence to show that at least the ranking or sign of effects is preserved. Could be done by swapping mCherry and YFP for instance in the plasmid.

We carried out the suggested swap for six uORF reporters (wildtype and mutant) and assayed the effects of the uORF on reporter protein expression using a fluorometer. The measurements were largely consistent, as shown in the new Figure 1—figure supplement 3 of the revised manuscript.

This is probably because of my background but I found the section on evolution really interesting and broadening the scope of the paper quite a lot. I wonder if the authors could not push further their analysis for instance by analyzing the polymorphism data available for 100s of strains in *S. cerevisiae*, for instance by looking at the frequency spectrum of mutations that would enhance the repressive effects of uORFs. Mutational bias could also be considered. With this part, it would also be interesting to know whether there are interactions between uORFs and the main ORFs because changes in magnitude or even signs of uORFs effects could be compensated by other regulatory changes, in a coevolution akin to what the same authors have reported on before regarding transcription and translation.

We thank the reviewer for these suggestions. While we agree this is an interesting avenue to explore, this is complicated by uncertainties in locating transcription start site positions in the other genome sequenced strains of *S. cerevisiae*. Thus, we feel it would be beyond the scope of this initial study and is better suited for future work.

Reviewer #2 (Recommendations for the authors):– They have not verified that a random sampling of reporter mRNAs generally have 5' ends corresponding to those predicted by using the ENO2 promoter in their library design. This is important because the position of uORFs from the 5' end is claimed to be an important determinant of uORF function, and represents one of the most novel findings of the study.

We agree with the reviewer. Our RNA-seq data were generated by 5’ end ligation. Because reverse transcription adds an untemplated “CNN” at m7G modified 5’ ends (Wulf et al., JBC 2019; Figure 3; PMID 31640989), our RNA-seq data can also be analyzed essentially as 5’ CAGE-seq data. We used this to identify the transcription start sites of our YFP reporter mRNAs. Reassuringly, we found that ~97% of our reporter mRNAs initiated at the designed transcription start site. We added the new Figure 1—figure supplement 2 describing these results and updated the main text (page 8), which are also described in more detail in the response to the editor’s essential revisions list above.

– They have not tested their conclusions about the importance of distance of the uORF from the mRNA cap on uORF function by follow-up experiments on a few individual repressive uORFs to examine whether increasing the distance of the uORF ATG from the cap by insertion of unstructured sequences (eg. CAA repeats) reduces inhibition by the uORF; or whether deleting sequences to move a uORF closer to the 5' end increases its inhibitory effect. This is important because the conclusion is based only on correlation analysis and hence could be influenced by other features of the mRNA that vary in parallel with distance of the uORF from the cap.– Regarding the distance from the cap effect, it seems possible that repression by an uORF would be diminished by a location further downstream from the cap owing to inclusion of other AUG or near-cognate start sites located upstream of the uORF, which would reduce translation initiation at the uORF of interest. This possibility should be addressed bioinformatically, and also by examining individual reporters in which all AUG or near-cognate triplets are removed upstream of the uORF in question to determine if this increases repression by the uORF without changing its position relative to the cap.

We thank the reviewer for these suggestions. We found that uORFs closer to the cap are more repressive. The reviewer rightly notes that this is based on a regression analysis of thousands of uORFs and had not been experimentally tested. To address this, we extended the 5’ UTRs upstream of two cap-proximal strong uORFs and deleted the 5’ UTRs upstream of two cap-distal mild uORFs. These mutations altered the effects of two of the uORFs. While one extension mutation did reduce uORF repression (as predicted by our ENR modeling), the deletion mutation resulted in a slightly less repressive uORF rather than a more repressive uORF (see new Figure 4—figure supplement 2). As detailed in our response to the reviewing editorial comments, we believe this may reflect additional effects of the mutations outside of changing uORF location (e.g. structural or transacting effects).

We further computationally investigated the possibility that the location effect results from having other upstream uORFs by restricting our correlation analysis to single AUG uORF transcript leaders (501 total; Figure 4—figure supplement 2). As observed with all uORFs, single AUG-uORFs are less repressive with increasing distance from the 5’ cap. The difference is significant with a one-tailed test (P = 0.04, P = 0.03), which is appropriate as our hypothesis is that the cap proximal group is more repressive (as opposed to simply “different”) than the cap distal groups. Stronger repression was also seen for uORFs that end closer to the CDS start codon in single AUG uORF transcript leaders, however this did not reach statistical significance (P = 0.6327 and P = 0.1884). Together, these analyses of single uORF transcript leaders are generally consistent with our interpretations from multiple uORF transcript leaders.

The reviewer also raised an important point regarding the activities of uORFs that are downstream of other uORFs. Our assay isolates repression of individual uORFs by comparing individual uORF AAG mutations to the wildtype UTR. Thus, we expect our measurements will be largely independent of upstream or downstream uORFs. To investigate this, we compared the strength of repression for first, second, third and fourth uORFs in multiple uORF transcript leaders. Strikingly, each had similar median levels of repression (Figure S4—figure supplement 2).

More broadly, the reviewer’s comments raise the important point that correlations from models are not necessarily causative. We agree and have revised the manuscript to better reflect the fact that the model identifies features that are predictive of uORF function (see pages 2, 7, 18, and 26). We believe this more accurately represents the results of the model.

– A point of major confusion: Most of the plots and supplementary tables list the reporter expression ratios as log2(AAG/WT); however, I believe that this incorrect and the log2(WT/AAG) ratios have been plotted instead, as the values for the majority of uORFs, which are repressive, are <0.

Thank you for pointing out this issue. It has been corrected in the revised figures.

­– p. 13: The sentence: "In general, repressor uORFs were less repressive in the upf1∆ strain, including 92% of AUG-uORFs and 59% of non-AUG uORFs." requires statistical analysis to bolster the claims, particularly for the small subset of repressive non-AUG uORFs

We apologize for omitting statistical analysis for this. We used a binomial exact test, with a null hypothesis that equal numbers of more repressive and less repressive uORFs would be seen in the upf1∆ strain. This results in p < 2.2 x 10^-16 for AUG uORFs (N=835) and p = 0.07135 (N = 136) for NCC uORFs. However, this analysis includes a large number of extremely weak NCC uORFs. After filtering out weak uORFs that reduced expression less than 15% in wildtype yeast, we found 93% of AUG (611 / 654, p < 2.2 x 10^-16) and 85% of NCC (22 / 26, p = 0.0005335) repressors were less repressive in the upf1∆ strain. We have added this to the revised manuscript (pg 12).

– p.14: The sentence "Consistent with this, we found median %NMD was higher for uORFs terminating with UGA (37.5%) than those terminating with UAA (35.3%) or UAG (33.2%) (Figure 3D)." is not justified, as the median values for the three stop codons in Figure 3D do not appear to differ significantly. Moreover, assuming it was valid, is there any way to rationalize it based on known termination efficiencies at the three stop codons?

The text notes that it has been reported that UGA is the least efficient stop codon (Bonetti et al., 1995). We also noted that previous studies showed UGAC is the least efficient context for termination. Thus, the variation we observe in NMD induction is consistent with the current model that NMD is induced by inefficient termination. However, as the reviewer correctly states, the difference in %NMD among the three stop codons was not statistically significant. We have updated the text to clarify that point (pg 14).

– Regarding the claim on p. 14 that "Unexpectedly, the location of the stop codon relative to the transcript leader cap was positively correlated with the %NMD, such that uORFs that terminate further from the cap were more likely to induce NMD than those that terminate adjacent to the cap (Figure 3F; R^2 = 0.065; P = 7.44x10^-8)", it should be noted that this is a very weak trend that explains only ~7% of the variance in %NMD values. As such, the sentence in the Discussion "We found the opposite relationship among uORF stop codons, such that stop codons were less likely to induce decay the closer they were located to the 5' cap" appears to be an overstatement. Also, isn't it possible that the trend can be explained as originating from uORF length effects shown in panel F, as longer uORFs will have greater cap to stop codon separations? (Note also that panels F and G were incorrectly cited in text.)

We agree, and revised the results and discussion to include the reviewer’s more nuanced interpretation and avoid overstating the correlation between uORF stop codon location and %NMD (pg 15 and 16). We also corrected citations to panels F and G.

– pp. 16-17 and Figure 4D: The description of the parameters included in the ENR model is wholly inadequate. How was Kozak context quantified and for what sequence interval around the AUG? How was uORF Start and Stop and CDS sequence conservation quantified (and for what species)? What sequence interval around the uORF Start was employed to calculate the deltaG folding energies, and how were the calculations made? How far downstream was %AU calculated. In general sufficient information has to provided in the legends or Methods to allow the analysis to be repeated in full by other workers. A related comment is that the authors cite no literature to justify their analyses of these different parameters which they seem to pull out of the hat, eg. why Pro and Gly codons?

We apologize for these omissions. We have included two new supplemental tables (Supplementary files 1j and 1k) with the features used for modeling and have revised the methods section extensively to describe feature sources, including citations (page 38). We measured relative Kozak scores using our YFP / mCherry reporter plasmid (new Figure 4—figure supplement 1). During revision, we found the Kozak scores used in the initial submission were mistakenly taken from a yeast strain with a different genetic background. We updated the Kozak scores using the wildtype (BY4741) strain and reran the ENR modeling for the resubmission. We also updated the %AU after uORF stop codon feature, as it was missing from uORFs that terminated close to the YFP start codon in the original submission. While the general results were the same (with location being as predictive as Kozak context), the ENR regression weights changed somewhat after modeling with these updated features, such that the wildtype and upf1∆ models were more similar than in the initial submission. We updated the Results section to reflect this.

– p.19 and Figure 5A: the claim that "In general, uORFs were more repressive when they were closer to the TSS" is not convincing, as many uORFs shown in blue in Figure 5A show the opposite trend; and there are only a few outliers that conform to the stated trend. No statistical analysis of the trend was provided. As such, it is an overinterpretation of the data to claim that these results independently support the importance of uORF distance from the TSS, indicated by the ENR analysis.

We agree that this required statistical evaluation. Of uORFS whose activity differed significantly by at least two-fold in the two transcript leaders, thirty-four were more repressive in the shorter transcript leader while twelve were more repressive in the longer transcript leader. We used the binomial exact test to evaluate this result under the null hypothesis that equal numbers of uORFs would be more repressive and less repressive in the context of the shorter transcript leader (P = 0.001641) and added this to the revised manuscript.

It also seems important to provide the YFP expression for the different leaders with WT and mutant uORFs, rather than just the WT/AGG ratios, in order to evaluate whether there are other sequences besides uORFs in the longer 5'UTRs that are affecting expression. This information should be added to Figure 5B in particular.

We agree that raw YFP values should be included with the study and added these to the supplemental table (Supplementary file 1g in the revised submission) and as bar graphs to Figure 5B as requested.

– p.20-21" Regarding the text: "For example, an S. cerevisiae oORF in SEC1 was 1.5 to 2-times more repressive than its *S. paradoxus* homolog, potentially owing to a deletion in *S. paradoxus* that results in an earlier stop codon that shortens the oORF. In another case, an *S. paradoxus* oORF in the AIM22 leader was approximately four-fold more repressive than its *S. cerevisiae* homolog, possibly due to the presence of more adenosines in its Kozak sequence." The words "potentially" and "possibly" in these sentences presumably reflect the fact that there are other sequence differences in the 5'UTRs between the two species that contribute to the differences in uORF function. These interpretations should be tested by mutational analysis of the reporters for these genes to determine whether shortening the oORF in S.c. SEC1 and improving the Kozak context of the S.c AIM22 oORF are sufficient to confer the altered repressive functions of the orthologous oORFs in S.p.

Although we suggested potential mechanisms for two species differences in uORF activity, other sequence or RNA structural features could certainly contribute. We cloned and tested additional reporters of the AIM22 uORF, swapping the Kozak sequences from the two species. The results were consistent with our suggestion that Kozak sequence differences contribute to these differences in activity (new Figure 5—figure supplement 1), however the Kozak sequence swap did not fully account for the difference. We revised the text to clarify that these are suggested potential contributors to uORF differences, and not necessarily the sole reasons underlying their differences. We believe dissecting the exact mechanisms underlying additional species differences in uORF magnitudes would be an interesting area for future work.

– Figure 6D and related text: It is not convincing that conserved uORFs have a statistically significant poorer AUG context compared to nonconserved uORFs, even if one focuses (as they do) on only the -3 position rather than calculating the context score for the entire sequence interval surrounding the AUG. This is being overinterpreted.

We agree with the reviewer, so we replaced the motif logo comparison with comparisons of Kozak context scores. This includes Wilcoxon rank-sum tests, which show that the lower Kozak scores in conserved uORFs are statistically significant.